# The impact of microRNAs on transcriptional heterogeneity and gene co-expression across single embryonic stem cells

Gennaro Gambardella[1,*], Annamaria Carissimo[1,*], Amy Chen[2,3], Luisa Cutillo[1], Tomasz J. Nowakowski[2], Diego di Bernardo[1,4] & Robert Blelloch[2,3]

MicroRNAs act posttranscriptionally to suppress multiple target genes within a cell population. To what extent this multi-target suppression occurs in individual cells and how it impacts transcriptional heterogeneity and gene co-expression remains unknown. Here we used single-cell sequencing combined with introduction of individual microRNAs. miR-294 and let-7c were introduced into otherwise microRNA-deficient Dgcr8 knockout mouse embryonic stem cells. Both microRNAs induce suppression and correlated expression of their respective gene targets. The two microRNAs had opposing effects on transcriptional heterogeneity within the cell population, with let-7c increasing and miR-294 decreasing the heterogeneity between cells. Furthermore, let-7c promotes, whereas miR-294 suppresses, the phasing of cell cycle genes. These results show at the individual cell level how a microRNA simultaneously has impacts on its many targets and how that in turn can influence a population of cells. The findings have important implications in the understanding of how microRNAs influence the co-expression of genes and pathways, and thus ultimately cell fate.

[1] Telethon Institute of Genetics and Medicine, Pozzuoli, 80078 Naples, Italy. [2] The Eli and Edythe Broad Center of Regeneration Medicine and Stem Cell Research, Center for Reproductive Sciences, University of California, San Francisco, San Francisco, California 94143, USA. [3] Department of Urology, University of California, San Francisco, San Francisco, California 94143, USA. [4] Department of Chemical, Materials and Industrial Engineering, University of Naples 'Federico II', 80125 Naples, Italy. * These authors contributed equally to this work. Correspondence and requests for materials should be addressed to D.d.B. (email: dibernardo@tigem.it) or to R.B. (email: Robert.Blelloch@ucsf.edu).

MicroRNAs (miRNAs) are short non-coding RNAs that arise through the biogenesis of long pri-miRNA transcripts[1]. Pri-miRNAs undergo an initial processing step by a complex consisting of the RNA-binding protein DGCR8 and the RNaseIII enzyme DROSHA, resulting in a hairpin structure called the pre-miRNA. The pre-miRNA is then processed by Dicer to form a short double-stranded RNA, a single strand of which is loaded into an Argonaute (Ago) to form the miRNA ribonucleoprotein effector complex. A predominance of miRNAs, called canonical miRNAs, follows this sequence of biogenesis events. A small number of non-canonical miRNAs bypass DGCR8-DROSHA processing, although these miRNAs are rare in comparison with the canonical miRNAs in mouse embryonic stem cells (mESCs)[2]. Thus, the deletion of the *Dgcr8* gene in mESCs results in essentially miRNA-deficient cells.

*Dgcr8*-null ESCs have a slower proliferation rate, with an extended G1 phase, relative to their wild–type (WT) counterparts when retained in self-renewal growth conditions (plus leukaemia inhibitory factor, LIF)[3]. *Dgcr8*-null ESCs are also unable to silence the pluripotency programme when placed under various differentiation conditions and thus remain locked in a self-renewing state[3]. Screens reintroducing individual miRNAs into these otherwise miRNA-deficient cells have been used to uncover the miRNAs responsible for these phenotypes. A family of miRNAs, called the ESC-enriched cell cycle regulating (ESCC) miRNAs can rescue the cell cycle defect in *Dgcr8*-null ESCs[4]. Another family of miRNAs, the let-7 family, can rescue the ability to silence the pluripotency network in *Dgcr8*-null ESCs[5]. miRNA families are defined by sharing a common seed sequence, a short six to eight nucleotide sequence near the 5′-end of the miRNA, which has been shown to play a central role in determining the downstream miRNA targets[6]. Thus, members of a miRNA family share largely overlapping messenger RNA targets.

Determining the targets of miRNAs and the impact of miRNAs on those targets is essential to understanding their mechanism of action[7]. To date, experimental identification and evaluation of target mRNAs, including those of the ESCC and let-7 family miRNAs[5,8,9], has been limited to cell population studies. Therefore, resulting findings are based on the averaged effect of the miRNAs across millions of cells. On the contrary, single-cell sequencing enables the measurement of miRNAs' effects within and across individual cells[10]. Single-cell sequencing can address several important roles of miRNAs that cannot be discerned from previous population studies. First, it can address transcriptional heterogeneity among cells in a population, thus assessing a miRNA's role in buffering gene expression variability at the genome-wide level[11,12]. Second, it can uncover coordinated fluctuations in gene expression across cells (that is, gene co-expression) that could reveal regulatory influences of a miRNA on specific sets of genes. Indeed, whether co-expression among direct and/or indirect miRNA targets across single cells occurs, and if so by which mechanisms, is still unclear. Here we use single-cell sequencing and reintroduction of individual miRNAs into Dgcr8[−/−] ESCs, to evaluate the impact of the ESCC and let-7 miRNAs on both of these phenomena.

## Results

### Reintroduction of miRNAs into miRNA-deficient cells.
To study the impact of one miRNA at a time without the complication of competing endogenous miRNA function, we introduced single miRNAs into otherwise miRNA-deficient Dgcr8 knockout mESCs (Dgcr8[−/−] mESCs)[3]. To suppress differentiation and minimize cell-state heterogeneity among mESC, cells were grown in the presence of LIF and inhibitors to GSKb and MEK (LIF + 2i) (see Methods), conditions

that maintain a stable pluripotent state by inhibiting autocrine differentiation cues[13–16]. To determine the efficiency of transfection, Dgcr8[−/−] cells were transfected with a fluorophore-conjugated control small RNA. Flow cytometry analysis of transfected cells showed a shift in fluorescence for the vast majority of cells (Supplementary Fig. 1a). Using identical conditions, cells were transfected with a representative let-7 family member let-7c and a representative ESCC family member miR-294. Resulting cells were dissociated and introduced into a Fluidigm C1 chip and analysed under a microscope, to confirm the presence of single cells in individual wells. We noticed that miR-294 induced an increase in cell size relative to Dgcr8[−/−] or let-7c-transfected cells. Quantitative analysis showed this effect across a vast majority of the cells (Supplementary Fig. 1b), confirming not only efficient introduction, but also function of the exogenously introduced miRNA.

### Single-cell sequencing of cells.
To understand how these miRNAs influence mRNA levels within and across cells, captured cells were lysed and mRNA was retrotranscribed into complementary DNA libraries. Sequencing adapters were introduced using a transposon-based fragmentation and sequenced on Illumina Ultra-High-Throughput sequencer. Sequencing reads were processed by a bioinformatics pipeline summarized in Fig. 1a (see Methods). Samples were filtered based on library depth, diversity and evidence of miRNA transfection (Supplementary Figs 2–4 and Methods). Evidence of miRNA transfection in each cell was determined by performing a gene set enrichment analysis (GSEA)[17], to assess the downregulation of previously defined targets (Supplementary Data set 1) of miR-294 and let-7c (ref. 5) (Fig. 1b). This analysis confirmed efficient transfection of cells with only a small number of cells having to be removed for lack of evidence for targets' downregulation (one for miR-294 and five for let-7c; Supplementary Fig. 4b). Of note, the GSEA analysis for miR-294 targets in miR-294-transfected Dgcr8[−/−] cells showed an identical enrichment score (ES) as that of WT cells, suggesting physiological function of the miRNA (Fig. 1b). In contrast, GSEA analysis of let-7c targets showed highly distinct ES in let-7c-transfected Dgcr8[−/−] cells relative to WT, which is expected as WT mESCs do not express let-7c (Fig. 1b). The numbers of samples remaining after each filtering step (see Methods) are summarized in Table 1. Genes were filtered based on a minimal average read count across samples (see Methods).

Principal component analysis (PCA) of the resulting matrix of samples and genes showed separation into three groups across PC1 consisting of let-7, Dgcr8[−/−] and WT/miR-294-transfected cells (Fig. 1c). Again, miR-294-transfected Dgcr8[−/−] cells overlapped with WT mESC in the PCA, suggesting not only physiological function of the exogenously introduced miRNA but also confirming the dominant role of the ESCC miRNA family in WT mESCs. The miR-294 and let-7c mimics led to shifts in opposite directions along the first principal component. GSEA using the loading values in PC1 revealed a strong enrichment for cell cycle-related gene sets (Supplementary Data set 2 and Methods). These single-cell data are consistent with previous population data, showing opposing roles for the ESCC and let-7c miRNAs in mESCs[5].

Let-7c induces the differentiation of Dgcr8[−/−] cells grown ESC media plus LIF[5]. However, 2i conditions blocks let-7c-induced differentiation[14]. We confirmed a lack of differentiation under any of the miRNA conditions, which were all performed in LIF + 2i, by evaluating six pluripotency markers (*Pou5f1*, *Nanog*, *Sall4*, *Esrrb*, *Klf4* and *Rex1*) and ten early differentiation markers (*Sox17*, *Brachyury*, *Fgf5*, *Sox1*, *Pax6*, *Grhl2*, *Mixl1*, *Gata4*, *Gata6*

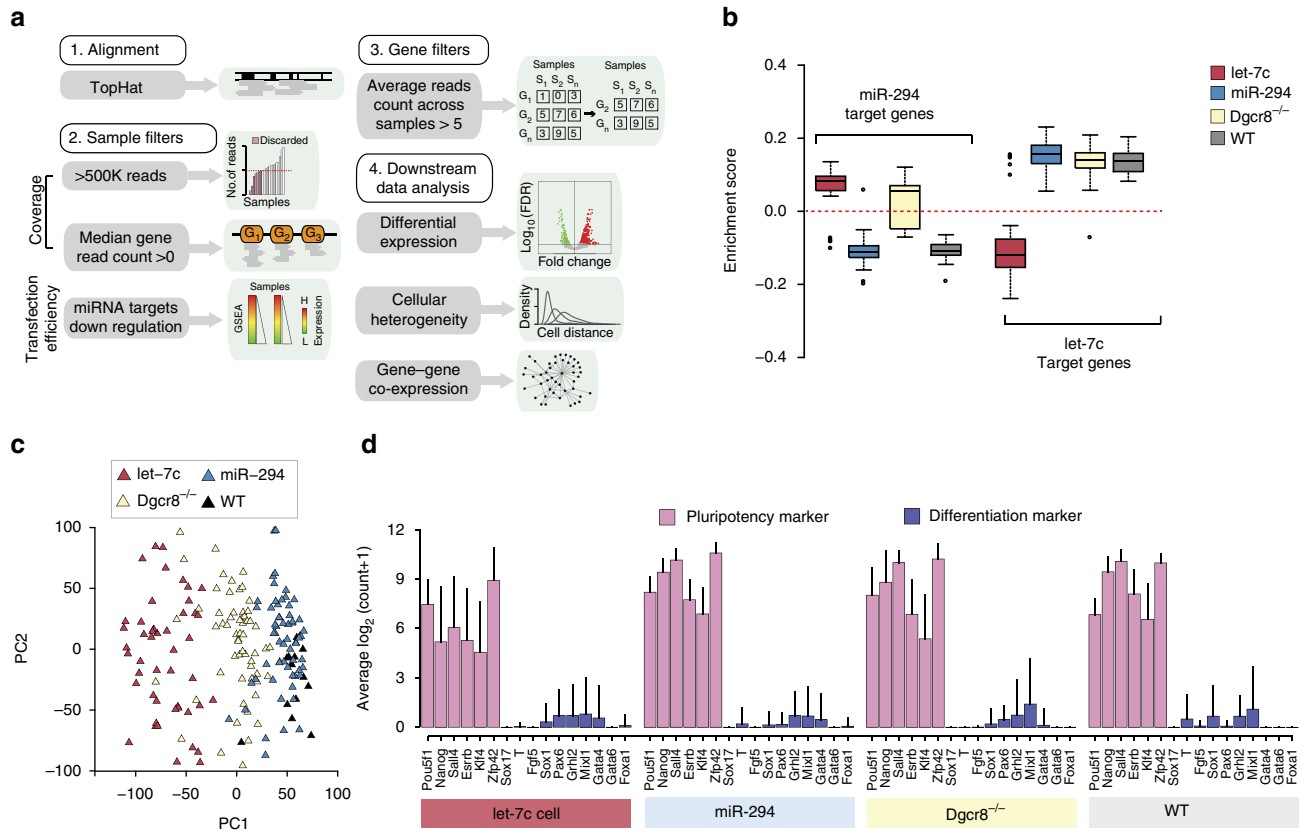

**Figure 1 | Single-cell sequencing of mESC transfected with either miR-294 or let-7c.** (**a**) Scheme of the bioinformatics and statistical pipeline for single-cell sequence analysis. Briefly, (i) reads are aligned with TopHat software and then (ii) cells with low coverage or with no evidence of miRNA transfection are removed and (iii) finally non-expressed genes are filtered out. Remaining genes are used (iv) for downstream analysis. Details provided in material and methods section. (**b**) ES distribution of miR-294 target genes across miRNA transfected, Dgcr8$^{-/-}$ or WT cells. (**c**) PCA analysis on filtered and normalized data showing individual cells colour coded by condition. The PCA analysis separates cells according to their condition. (**d**) Average expression of 16 pluripotency/differentiation markers in each condition.

| Table 1 \| Number of samples under each condition following each filter. | | | |
|---|---|---|---|
| | **>500,000 reads** | **Median read count >0** | **GSEA filter** |
| Dgcr8$^{-/-}$ | 61 | 60 | 60 |
| WT | 17 | 16 | 16 |
| let-7c | 56 | 53 | 48 |
| miR-294 | 61 | 59 | 58 |
| GSEA, gene set enrichment analysis; WT, wild type. | | | |

and *Foxa1*). None of these markers showed a significant change across the four conditions (Fig. 1d). Therefore, the downstream transcriptional effects of these miRNAs could not be ascribed to secondary effects associated with differentiation.

**Differential expression across conditions.** Next we performed differential gene expression analysis, to determine whether single-cell data could recapitulate bulk population findings across differing conditions. Previously, Affymetrix array studies had been performed on Dgcr8$^{-/-}$ ESCs plus or minus miR-294 or let-7c, revealing hundreds of downregulated genes containing a seed match corresponding to the corresponding to transfected miRNA, which were identified as probable targets of that miRNA[5]. Analysis of the single-cell data showed a very similar effect to that of the bulk studies (Fig. 2a and Supplementary Data set 3). Not a single target from the bulk population study showed

a significant opposite effect in the single-cell data (Fig. 2a). However, fewer targets were found to be significantly downregulated following miRNA introduction in single-cell data, which probably reflects the reduced sensitivity of this method. As expected, the miR-294 targets were similarly downregulated in WT mESCs, where the ESCC miRNAs make up a majority of the miRNA pool[2,18]. In contrast, the let-7c targets were not, which is also expected, as let-7 family members are not expressed in ESCs. The previous bulk population study had shown that miR-294 and let-7c had opposing effects on the Myc pathway. Here we asked whether the Myc pathway and/or alternative pathways could distinguish individual cells in each of the four conditions. Machine learning was used to determine the predictive power of 50 hallmarks gene sets annotated in the MsigDb database that describe genes with coordinated expression levels in a variety of tissues involved in specific biological processes[19] (Supplementary Fig. 5 and Methods). This analysis uncovered the Myc pathway as having the highest predictive power, identifying miR-294 and let-7c cells with a >90% accuracy (Fig. 2b). Therefore, our single-cell data, when considered as a group, correlate well with previous bulk population studies. In contrast to these studies though, the single-cell data enabled us to next ask how the miRNAs have an impact on transcript levels within and across individual cells.

**Effects of miRNAs on transcriptional heterogeneity.** Single-cell transcriptome data enables the analysis of transcriptional heterogeneity within a population of cells[14]. WT ESCs grown in

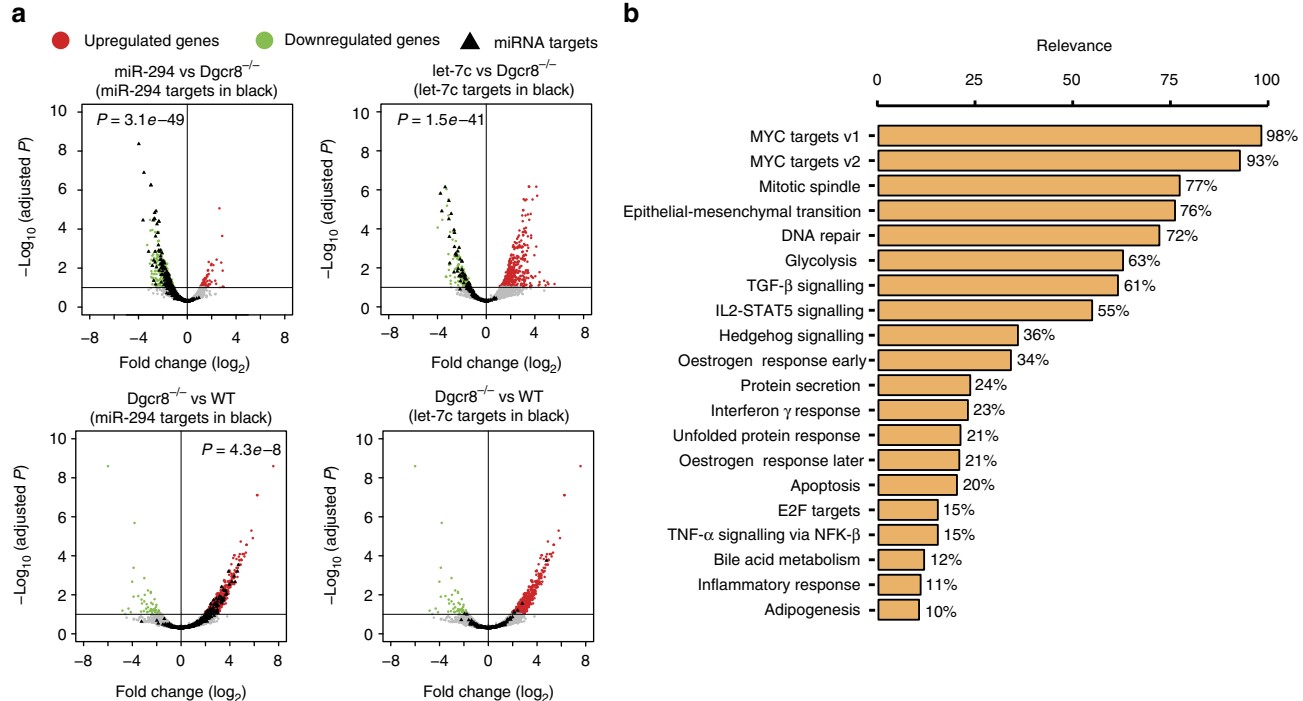

**Figure 2 | Differential gene expression of mESC transfected with either miR294 or let-7c.** (**a**) Volcano plots showing expression fold change of mean across cells in each condition on *x* axis and -$\log_{10}$(FDR) on *y* axis. Individual cells within a condition were treated as repeats for differential expression analysis. Significantly differentially expressed miRNA targets (FDR <10%) for miR-294 and let-7c identified in previous population-based array experiments are highlighted as black triangles and *P*-values for the enrichment analysis is shown in upper left corner. miR-294 and let-7c targets are similarly highlighted in Dgcr8$^{-/-}$ versus WT mESC comparison. Conversely to let-7c, miR-294 is broadly expressed in mESC and its targets are significantly upregulated in miRNA-deficient Dgcr8$^{-/-}$ cells versus WT mESC. (**b**) Identification of discriminant processes between miR-294 and let-7c-transfected cells. Pathways are sorted according to their relevance by mean of the frequency they were selected by the RFE method to correctly assign a cell to its category. Only pathways selected at least in 10% of 1,000 trials are shown.

LIF+2i are highly transcriptionally homogenous, whereas miRNA-deficient cells show an increase in heterogeneity compared with their WT counterparts[14]. The effect of individual miRNAs on the transcriptional heterogeneity of cell populations remains unknown. To address this question, we evaluated all potential pairwise correlations between the transcriptional profiles of single cells within each of the three cell populations: Dgcr8$^{-/-}$ cells, miR-294-transfected Dgcr8$^{-/-}$ cells and let-7c-transfected Dgcr8$^{-/-}$ cells (Fig. 3a). Compared with the Dgcr8$^{-/-}$ cells, the addition of miR-294 increased the correlation between cells (one-tailed Mann–Whitney test $P < 2.2e-16$), implying a reduction in cell-to-cell transcriptional variability (Fig. 3a, inset in upper panel). In contrast, compared with Dgcr8$^{-/-}$ cells, the introduction of let-7c decreased the correlation between cells (one-tailed Mann–Whitney test $P < 2.2e-16$), implying an increase in cell-to-cell transcriptional variability (Fig. 3a, inset in upper panel). These effects were even greater when specifically focusing on the cell-to-cell transcriptional variability of pluripotency regulators (Fig. 3a, lower panel). Furthermore, these effects were largely driven by the highly expressed genes as intermediate and lowly expressed genes showed little correlation between cells in any condition, likely due to poor read coverage for these genes with the single cell sequencing method (Supplementary Fig. 6). These results show that individual miRNAs can either reduce or enhance transcriptional heterogeneity across cells.

Although transfection of let-7c resulted in increased variation between cells of the population as a whole, hierarchical clustering of single-cell profiles showed the formation of clusters of cells where transcriptionally heterogeneity appears to be lower, effectively structuring the cell population into distinct subpopulations (Fig. 3b). In contrast, miR-294-transfected Dgcr8$^{-/-}$ cells did not form subpopulations and as a whole were highly transcriptionally homogenous. Non-transfected Dgcr8$^{-/-}$ cells also formed subpopulations, but were less discrete than those seen with the addition of let-7c (Fig. 3b). Inter-cluster distance analysis confirmed the greater distinctness of subpopulations within the let-7c-transfected versus non-transfected Dgcr8$^{-/-}$ cells (Fig. 3c, two tailed Mann–Whitney test $P = 1.94e-8$). PCA analysis confirmed that the let-7c subpopulations could be separated on the first principal component, whereas the Dgcr8$^{-/-}$ subpopulations separated on the third principal component, albeit less distinctively, in agreement with the inter-cluster distance analysis (Supplementary Fig. 7 and Supplementary Data set 2). These findings show that the increase in transcriptional heterogeneity seen in let-7c-transfected cells is largely the result of the production of subpopulations rather that increased stochastic noise between cells.

**Source of transcriptional heterogeneity.** Next, we asked what genes/pathways are driving the formation of these subpopulations. We thus performed GSEA for the 50 hallmark gene sets within each subpopulation of let-7c-transfected cells and non-transfected Dgcr8$^{-/-}$ cells. We thus obtained an ES for each gene set in each cell in each subpopulation. We then performed an analysis of variance (ANOVA) test, to determine gene sets with an ES significantly different across the subpopulations within each condition. This analysis uncovered 13 and 9 significant gene sets (false discovery rate (FDR) <10%) driving the formation of let-7c and Dgcr8$^{-/-}$ subpopulations, respectively (Fig. 3d and Supplementary Data set 4). Interestingly, there was a

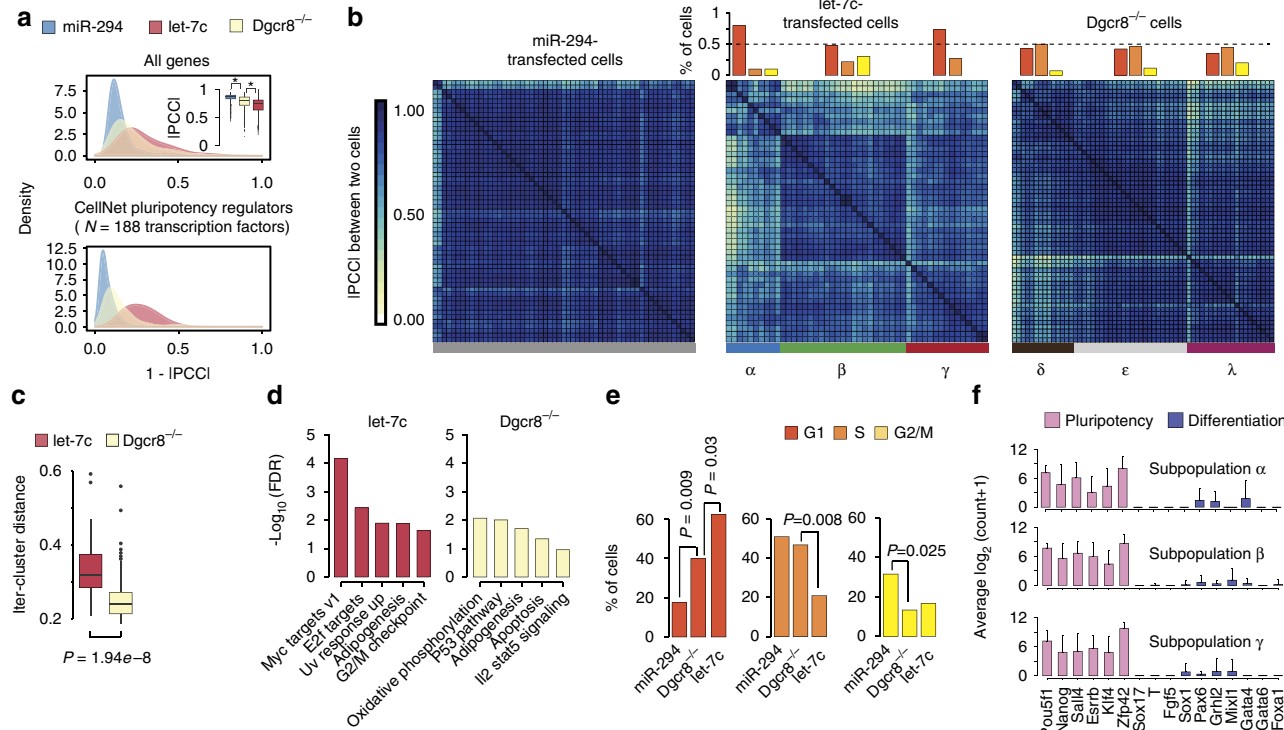

**Figure 3 | Cell–cell correlations within Dgcr8$^{-/-}$ and Dgcr8$^{-/-}$ transfected with either miR294 or let-7c.** (**a**) Density plots of distances (|1–|PCC|) between pairs of cells within each condition. Top plot is for all genes, whereas bottom plots is for CellNet pluripotency regulators as defined in ref. 14. Top plot inset shows PCC distribution between pairs of cells within each condition. It is worth noting that miR-294 increases homogeneity when considering all genes or pluripotency genes. (**b**) Heatmaps showing all within condition cell–cell PCC using all genes. Cells are ordered by hierarchical clustering and subpopulation of cells identified with Dynamic tree cut method (see Methods). Above each heatmaps the percentage of cells in each of the cell cycle phases is reported for each identified sub-population of cells as defined by Cyclone. (**c**) Inter-cluster distance distribution between clusters (that is, subpopulations) of cells shown in **b**. Inter-cluster distance for a cell *x* was defined as its average distance from all the other cells, expect the ones in the same subpopulation as cell *x*. Let-7c inter-cluster distance is significantly higher than Dgcr8$^{-/-}$ inter-cluster distance (two tailed Wilcoxon test $P = 1.94e - 8$), meaning that let-7c is producing more distinct and better separated sub-populations of cell compared to the ones identified in Dgcr8$^{-/-}$ cells. (**d**) Identification of pathways driving the formation of subpopultions of cells within let-7c or Dgcr8 knockdown conditions. Pathways are sorted according the corrected *P*-value (that is, FDR) returned by the ANOVA. Only top five significant pathways are shown. (**e**) Percentage of cells in each of the cell cycle phases for miRNA-transfected and Dgcr8$^{-/-}$-deficient cells. Exact Fisher's test is used to the compare number of cells in each cell cycle stage between miRNA-transfected versus Dgcr8-knockdown cells. (**f**) Average expression of six pluripotency and ten differentiation markers across the three identified subpopulations of let-7c-transfected cells.

highly significant enrichment for gene sets related to cell cycle in let-7c cells including E2f targets and G2/M checkpoint (Fig. 3d and Supplementary Data set 4).

These data suggested that the transcriptional heterogeneity among cells observed following let-7c transfection may be associated with the regulation of cell cycle. Therefore, we used the single-cell data to ask how the cells were distributed across the cell cycle phases in each condition. To do this, we used a recently developed bioinformatics tool, Cyclone, which can predict cell cycle phase based on transcriptional markers, which were uncovered in mouse ESCs and thus well suited for our data[20] (see Methods). Cyclone showed distinct cell cycle distributions between miR-294-transfected, let-7c-transfected and Dgcr8$^{-/-}$ cells (Fig. 3e). Consistent with previous data using DNA content to measure the cell cycle phases[4,21], miR-294 decreased (exact Fisher's test $P = 0.009$), whereas let-7c increased (exact Fisher's test $P = 0.03$) the number of cells in G1 relative to the Dgcr8$^{-/-}$ control cells. Let-7c also decreased the number of cells in S phase (exact Fisher's test $P = 0.008$), whereas miR-294 increased the number of cells in G2/M (exact Fisher's test $P = 0.025$). We next applied the Cyclone tool to the subpopulations identified in the let-7c-transfected Dgcr8$^{-/-}$ cells. The first subpopulation (alpha) of let-7c-transfected cells consisted of mostly G1 cells, the

second subpopulation (beta) had cells in all three phases and the third subpopulation (gamma) consisted of G1 and S cells (Fig. 3b, top middle panel). In contrast, all three subpopulations of Dgcr8$^{-/-}$ cells showed cell cycle heterogeneity (Fig. 3b, top right panel). Together, these data show the changes in transcriptional heterogeneity, especially with let-7c-transfected cells, are associated with changes in distribution of cells across the different phases of cell cycle. This change could not be ascribed to the induction of differentiation, as analysis of the let-7c-transfected subpopulations showed no change in the expression of pluripotency and differentiation markers (Fig. 3f).

**Impact of miRNAs on gene co-expression.** The changes in cell cycle distribution alone cannot explain the changes in transcriptional heterogeneity. That is, although the cell cycle distribution changed, each condition still had some fraction of cells in each phase of the cell cycle, yet it was only in the let-7c-transfected cells where distinct transcriptional subpopulations formed around the different phases of the cell cycle, especially G1 and G1/S. Therefore, we next asked how let-7c and miR-294 affect co-expression of genes. Gene co-expression refers to how similar the expression of two (or more) genes is in terms

of correlated changes in their expression across single cells, such that the level of one gene in a cell is predictive of the level of the other gene in the same cell. The induction of gene co-expression among genes that vary across cell cycle could explain the formation of the distinct transcriptionally homogenous subpopulations of cells observed in let-7c-transfected cells.

Co-expression among a set of genes can be quantified by computing the pair-wise correlation coefficients between all possible gene pairs in the set and then performing a summary statistic on those correlation coefficients[22]. An alternative and more robust approach is to compute the Rényi multi-information (RMI), where the co-expression of all the genes in a set is quantified at once in a 'set-wise' manner, as opposed to 'pair-wise'[23]. We applied both pair-wise correlation and set-wise RMI approaches to evaluate the effects of miR-294 and let-7c on their respective high-confidence target sets (Supplementary Data set 5, Supplementary Fig. 8 and Methods). The pair-wise correlation analysis showed that, within each condition, the introduced miRNA increased the co-expression of its targets relative to the other miRNA's target set (Supplementary Fig. 9).

This effect was even stronger when using the set-wise correlation analysis (RMI; Fig. 4a,b, upper panels). The significance of these effects was confirmed by performing permutation tests on randomly selected genes of equivalent expression levels (Fig. 4a,b, lower panels). Furthermore, this finding was extended to a larger set of predicted targets of each miRNA (Fig. 4c and Supplementary Fig. 10). These data show that both miR-294 and let-7c increase the co-expression of their targets, despite the expression of their targets being downregulated (Figs 1b and 2a).

Given the impact of these miRNAs on cell cycle, we next asked how they influence the co-expression of a set of 36 well-annotated cell cycle phase genes (Supplementary Data set 6) including one direct target (*Cdkn1a*) of miR-294 and two targets (*Ccnf* and *Rrm2*) of let-7c (high-confidence targets in Supplementary Data set 5). Previous single-cell expression analysis had shown that these genes are well correlated according to cell cycle phase in somatic cells, but not ESCs[24]. Analysis of our data showed a similar lack of correlation between genes within each cell cycle phase among the miR-294-transfected cells (Fig. 4d). This effect

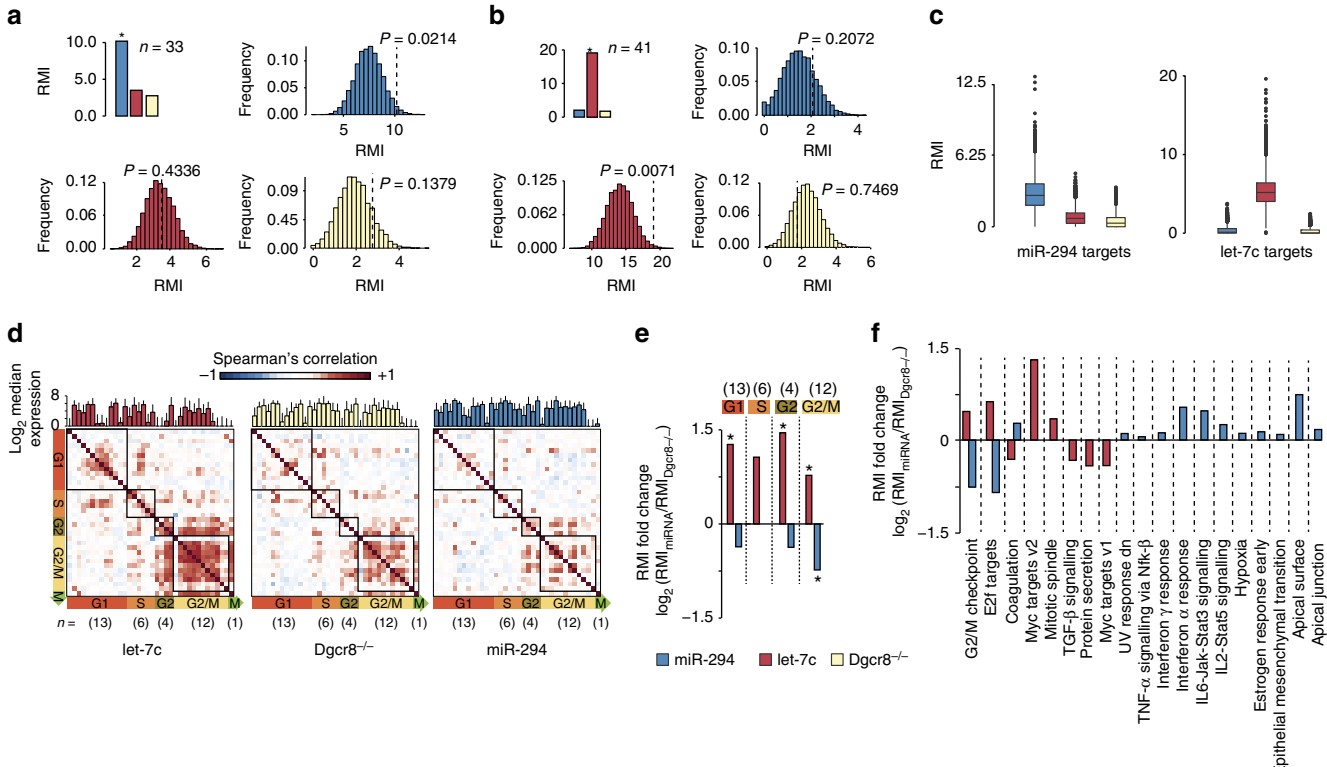

**Figure 4 | Gene co-expression across cells in Dgcr8$^{-/-}$ and Dgcr8$^{-/-}$ transfected with either miR294 or let7c.** (**a**) RMI for a set 33 of high-confidence miR294 targets in miR294-transfected (blue), let-7c-transfected (red) and Dgcr8$^{-/-}$ cells (yellow). Histograms show permutation test for random sets of genes expressed at the same levels as the targets under each condition. Green dotted line represents RMI of the targets themselves with associated p-values. (**b**) Same as **a**, except for a set of 41 high-confidence let-7c targets. (**c**) RMI distribution for each of the three conditions using larger sets of miRNA targets (defined as down with addition of miRNA and predicted by Targetscan, miRanda-miRSVR or previous population array data). As power of RMI is reduced by the larger size of these target sets, a distribution of RMIs was computed by randomly extracting a subset of 10 genes with replacement 10,000 times from the corresponding list of targets. (**d**) Spearman's correlations among 36 cell cycle-regulated transcripts in miRNAs transfected and Dgcr8$^{-/-}$ cells show an increase of cell cycle-dependent transcription in let-7c-transfected cells. Genes are grouped by cell cycle phases (squares) and ordered in the same way across conditions. Above each heatmap the average expression of each gene and its standard deviation is reported. (**e**) Differential RMI among 36 cell cycle-regulated transcripts (Supplementary Data set 7) in miRNAs transfected versus Dgcr8$^{-/-}$ cells show an increase of cell-cycle-dependent transcription in let-7c transfected cells. Genes are grouped by cell cycle phases and their RMI value was compared in miRNA transfected cells versus Dgcr8$^{-/-}$ cells. The number of genes used for each cell cycle phase is reported in the upper part of the plot. Significant changes of RMI determined by permutation test are indicated with asterisks (let-7c G1/S $P = 0.026$, G2 $P = 0.04$, G2/M $P = 3e - 3$; miR-294 G2/M $P = 0.016$). (**f**) Differential RMI among hallmark gene sets from MSigDb in miRNAs transfected versus Dgcr8$^{-/-}$ cells. For each hallmark gene set we computed RMI across the three conditions (miR-294, let-7c and Dgcr8$^{-/-}$) and then computed the RMI fold-change (miR-294 versus Dgcr8$^{-/-}$ and let-7c versus Dgcr8$^{-/-}$) and its significance determined by permutation test. Only genes set with significant changes are shown.

was enhanced relative to the miRNA-deficient Dgcr8$^{-/-}$ cells, suggesting that miR-294 plays a central role in this phenomenon of lost correlation among cell cycle phase genes in ESCs. In stark contrast, let-7c caused an increased correlation among genes of each phase of the cell cycle relative to the miRNA-deficient Dgcr8$^{-/-}$ cells (Fig. 4d). These effects were also confirmed by RMI analysis (Supplementary Fig. 11) with 10,000 permutations to assess significance (Fig. 4e).

These changes in correlation among cell cycle genes cannot be simply ascribed to changes in fraction of cells in each phase of the cell cycle: miR-294 increased the number of cells in G2/M, yet decreased the co-expression of the G2/M phase genes, whereas let-7c increased the number of cells in G1 and increased the co-expression of both G1 phase genes and G2/M phase genes (Figs 3e and 4d,e). One way in which miR-294 could lead to a loss of co-expression of cell cycle phase genes is through (direct or indirect) downregulation of their expression, leading to a loss of correlation. We quantified the gene expression of the 36 cell cycle phase genes across the three conditions (let-7c, Dgcr8$^{-/-}$ and miR-294) and found genes to be similarly expressed (Fig. 4d, upper panels). These results show that let-7c and miR-294 have opposite effect on the co-expression of cell cycle phase genes that are not simply secondary to their effects on the distribution of cells across the cell cycle or to overall changes in gene expression.

To detect other sets of genes that become co-expressed in the presence of either miR294 or let-7c, we applied RMI analysis (Supplementary Fig. 11) to the 50 hallmark gene sets (see Methods). For each hallmark gene set, we computed RMI across the three conditions (miR-294, let-7c and Dgcr8$^{-/-}$) and then computed the RMI fold change (miR-294 versus Dgcr8$^{-/-}$ and let-7c versus Dgcr8$^{-/-}$) and its significance (FDR < 10%) with a permutation test (see Methods). In agreement with our results on cell-cycle genes, miR-294 reduced the co-expression of genes in hallmark gene sets related to G2/M checkpoint and E2f targets (Fig. 4f and Supplementary Data set 7). In contrast, let-7c increased co-expression of genes in these hallmark gene sets and mitotic spindle (Fig. 4f). MiR-294 did increase co-expression of genes in other gene sets relevant to ESC biology such as Stat3 signalling and epithelial–mesenchymal transition. These effects cannot simply be explained by enrichment of direct targets in these pathways, as the effects were present even after removing known targets of each miRNA from the gene sets (Supplementary Figs 12 and 13).

## Discussion

In this study, we provide the first genome-wide analysis of how a single miRNA impacts the transcriptome of individual cells. We use single-cell transcriptome sequencing together with the reintroduction of individual miRNAs in an otherwise miRNA-deficient cellular background. The use of miRNA-deficient cells removes the potential noise created by the impact of competition of miRNAs. That is, the addition of small RNA can compete out endogenous miRNAs for binding to Ago and thus formation of a functional silencing complex[25]. Similarly, removal of an individual miRNA can allow other endogenous miRNAs to fill Ago enhancing their function. Thus, the Dgcr8-null ESC model essentially provides a clean slate to study the impact of individual miRNAs on the cellular transcriptome. The unnaturally high levels of the exogenous introduced small RNAs into the miRNA-deficient background could in theory lead to non-physiological targeting and downstream impacts of the miRNA. However, that is unlikely a concern with either of the two miRNAs studied here. miR-294 and let-7c are both members of large families of miRNAs that have been shown to already exist at saturating levels in their respective WT cell contexts[26]. Importantly, we show at

the single-cell level, the introduction of exogenous miR-294 alters the transcriptome of Dgcr8-null cells to one very similar to WT. Furthermore, it reduces its known targets to a similar degree. These findings are consistent with fact that the ESCC miRNAs constitute over 50% of the total miRNA pool in ESCs[2,18]. Let-7 similarly constitutes a high fraction of the miRNA population in downstream differentiated somatic lineages[18].

Our approach enabled us to study the impact of the individual miRNAs on cell-to-cell transcriptome variability and gene co-expression across cells. Previously, it has been shown by single-cell sequencing that the Dgcr8-null ESCs show greater transcriptional heterogeneity than their WT counterparts[14]. However, it was unclear how individual miRNAs would influence transcriptional heterogeneity between cells. Here we show very different effects of the two miRNAs, miR-294 and let-7. The introduction of miR-294 into the Dgcr8$^{-/-}$ induces a highly transcriptionally homogenous population, essentially the same as described for WT ESCs grown in similar culture conditions[14]. In contrast, let-7 increased transcriptional heterogeneity. However, a closer evaluation of that heterogeneity showed that this increase could be largely ascribed to the formation of subpopulations of cells. Analysis of the underlying genes driving the formation of these subpopulation uncovered a strong enrichment for cell cycle genes, especially G1 phase genes mostly included in the E2f target hallmark gene set. Interestingly, previous work had shown that the ESCC and let-7 miRNAs have opposing effects on the fraction of ESCs in the G1 phase. This result was confirmed here using the Cyclone tool on our single cell profiles to assign cells to the proper cell cycle phase.

Previous single-cell sequencing showed poor correlation among cell cycle phase genes in WT ESCs, but strong correlation in somatic K562 cells, suggesting a highly distinct structure of cell cycle gene expression in somatic versus ESCs[24]. We found that the introduction of miR-294 and let-7 induced distinct outcomes on the correlation of the cell cycle phase genes used in the previous study. Specifically, miR-294 induced a loss of gene co-expression very similar to WT ESCs, whereas let-7 induced a gain in gene co-expression similar to that of the K562 cells. Interestingly, let-7 is highly expressed in K562 cells[27]. This effect on cell cycle phase gene co-expression cannot be solely ascribed to changes in number of cells in each phase of the cell cycle. Indeed, miR-294 increased the number of cells in G2/M, but reduced the co-expression of G2/M cell cycle phase genes, whereas let-7c increased co-expression of G2/M genes, despite exhibiting a much smaller number of cells in G2/M. Let-7 normally increases as ESCs differentiate down somatic lineages and the introduction of let-7 in Dgcr8$^{-/-}$ ESCs grown in LIF alone induces differentiation of ESCs. Therefore, it is plausible that the let-7 effect on co-expression of cell cycle phase genes could be an indirect result of its impact on differentiation. However, in the studies presented here, the cells are grown in LIF + 2i conditions and we show that let-7 does not induce differentiation under these conditions. This finding is consistent with previous work showing that under 2i + LIF conditions, let-7 may actually reduce differentiation[14]. Therefore, the effect of let-7 on cell cycle gene phasing is likely to be due to be a more direct effect on the cell cycle pathways, although the mechanism remains to be revealed.

Single-cell sequencing following introduction of a miRNA in an otherwise miRNA-deficient background also allowed us to also study the impact of each miRNA on its downstream targets. Although it is well appreciated that miRNAs can bind and suppress multiple targets it is unclear whether it does so equally in each cell as all previous studies measured a population of cells. Our GSEA analysis confirmed that both miR-294 and let-7c

induce downregulation of their targets in individual cells. Another open question is whether miRNAs induce expression covariation among their targets, similar to that seen with transcription factors[28–30]. Using both pair-wise and set-wise correlation measures, we found that both let-7 and miR-294 led to significant increases in co-expression among their respective targets across individual cells. That is, the targets varied together from one cell to the next. The reason for induction of co-expression is unclear. It is unlikely to be due to varying amounts of the miRNAs as both exogenously introduced levels in Dgcr8$^{-/-}$ cells and endogenous levels in WT ESCs (for miR-294) are very high and likely to be saturating[26]. One possible explanation is that there is a rate-limiting factor in each cell that determines how much the targets can be degraded (for example, varying levels of Ago from one cell to the next). This can lead to the development of 'waiting lines' for biochemical processing, as recently shown in the case of two unrelated proteins tagged for degradation by the proteasome[31]. A similar or related mechanism may be occurring with miRNAs, although it requires further investigations.

In summary, we have used single-cell sequencing together with individual miRNA manipulation, to assess the impact of miRNAs on shaping the transcriptional profiles in individual cells. We find that miR-294 and let-7 are able to induce co-expression of their target genes, while having opposing effects on the co-expression of cell cycle phase genes and cellular heterogeneity in ESCs (Supplementary Fig. 14). Our work more broadly provides an approach to better understand the impact of miRNAs on their targets and ultimately on the biology of a population of cells.

## Methods

**Cell culture and transfections.** Dgcr8-knockout and WT parental cells were previously derived in the Blelloch lab (ref. 3) and can be obtained from Novus biologicals. Cells were maintained in knockout DMEM medium (Invitrogen) supplemented with 15% fetal bovine serum, LIF and 2i (PD0325901 and CHIR99021) as per standard techniques[32]. Cells were transfected with miRNA mimics (MIRIDIAN, GE Dharmacon) using Dharmafect 1 as previously described[33]. A fluorescence tagged (Dy547) control mimic was used to evaluate transfection efficiency (Supplementary Fig. 1a). Flow cytometry (LSRII) and microscopic analysis of transfected cells was performed 24 h post transfection. Mock-transfected control was treated under identical conditions, with the exception of the absence of the mimic.

**Analysis of cell size.** Images of Dgcr8$^{-/-}$ and miR-294 cells stained with the LIVE/DEAD Cell Staining Solution size were processed in ImageJ[34]. Colour images were converted to greyscale, then an automated threshold was set and the plugin Analyze particles was used to measure cell area in the segmented images (Supplementary Fig. 1b).

**Single-cell sequencing.** Twenty-four hours post transfection, cells were loaded and single-cell libraries prepared following the Fluidigm C1 protocol (version PN 100-7168 G1). One C1 integrated fluidic circuit (small cell size, PN 100-5759) was used for each condition. LIVE/DEAD Cell Staining solution (ThermoFisher) was included. Each cell capture site was visually inspected and photographed on an inverted fluorescent microscope. Each site was manually scored for cell number and viability. All multi-cell and dead cell sites were discarded from further downstream analysis. Sequencing libraries were produced using the Illumina Nextera XT DNA Sample Preparation kit following the manufacturer's guidelines. Libraries were quality control tested on a combination of Agilent Bioanalyzer and Tape Station and quantified using Qubit (ThermoFisher). Libraries were sequenced on Illumina HiSeq 2500 Ultra-High-Throughput Sequencing System.

Given the goal of identifying cellular gene networks, we reasoned it would be important to have significant sequencing depth of all genes evaluated. Initial sequencing depths ranged from < 1,000 to over 2.5 million reads (Supplementary Fig. 2a). Plotting per cent of genes with at least 10 reads to read depth showed a plateau at ~500,000 reads (Supplementary Fig. 2b). Therefore, samples under 500,000 and above 100,000 reads were resequenced to insure statistical robustness of downstream network analysis. Resequencing showed no evidence of batch effect as evidenced by PCA and thus were directly added to reads of first sequencing run resulting in a majority of samples ranging from 900,000 to 2 million reads (Supplementary Fig. 2c,d).

**Read mapping and normalization.** Reads were aligned and assigned to Gencode Mouse released M4 transcripts and genes by using RSEM version 1.2.19 with standard parameters[35]. Similar to previous studies, there was not a strong correlation between gene length and number of reads (Supplementary Fig. 3)[24]. Reads were normalized for sequencing depth using the method developed in DESeq[36].

**Sample and gene filters.** To reduce extraneous noise that would negatively impact network analyses, we established sample and gene filters based on number and diversity of reads, as well as evidence of introduced miRNA function. First, we evaluated library complexity based on median and spread of read counts across genes. A small number of cells showed a median equal to 0 with a majority of counts coming from a small number of genes (Supplementary Fig. 4a). These cells were removed from further analysis. Even though transfection efficiencies were very high, it is likely to be that a small number of cells did not receive the introduced miRNAs. To identify these cells, gene set enrichment analysis[17] was performed using the respective targets for miR-294 and let-7c identified in Melton et al.[5] Cells receiving the miRNA are expected to have an overall reduction in enrichment for the corresponding miRNA targets. One cell among the miR-294 and five cells from the let7-c cohort were removed, as unlike most of their cohorts they did show a negative enrichment (Supplementary Fig. 4b). Following above filters, 16 WT, 60 Dgcr8$^{-/-}$, 58 miR294-transfected and 48 let7c-transfected samples remained (Table 1). To remove noise from lowly expressed genes, we performed a Kolmogorov–Smirnov test, to measure the distance between samples that were re-sequenced along different cutoffs for minimal number of reads per gene. Minimal distance was found at a cutoff of five reads and thus all genes with average read count of less than five reads across samples were removed from further analysis.

**PCA and differential expression.** PCA analysis was performed on the normalized counts using the *prcomp* function of the 'stats' package in R environment. Figure 1c shows PCA based on 11,182 genes that passed filtering by average read counts greater than five reads across samples, whereas Supplementary Fig. 2c shows PCA based on 24,142 genes having at least one read in at least on sample. Supplementary Fig. 7 shows PCA based on the same genes as in Fig. 1c, but performed on either let-7c or Dgcr8$^{-/-}$ cells. Differentially expressed genes among conditions (Let-7c versus Dgcr8$^{-/-}$, miR-294 versus Dgcr8$^{-/-}$ and Dgcr8$^{-/-}$ versus WT) were detected using a Bayesian approach to single-cell differential expression analysis method[37]. To compare expression of a given gene between two groups, we used maximum likelihood estimate for the expression fold change on log2 scale. P-value was corrected for multiple hypothesis testing using Holm procedure. The differentially expressed genes (adjusted P-value < 0.1) are listed in Supplementary Data set 3, whereas the miRNA target genes from previous population study used in Fig. 2a are reported in Supplementary Data set 1.

**Hallmark gene sets.** Hallmark pathways for human species were first downloaded from the MSigDb repository[17], version 5, and then converted to *Mus musculus* using HomoloGene database (release version 68) (ftp://ftp.ncbi.nih.gov/pub/HomoloGene/build68/).

**Recursive feature elimination.** A machine-learning approach based on recursive feature elimination (RFE) and support vector machines (SVMs) was used to identify the pathways that were best at discriminating miR-294- and let-7c-transfected cells by their gene expression profiles (GEPs).

The RFE algorithm couples feature selection with SVMs[38]. Feature selection was used, to identify a minimal informative set of features, discarding uninformative or redundant ones. For SVMs with a linear kernel, as the ones used in this manuscript, RFE uses $||w||^2$ as a ranking criterion for the importance of a feature. The features with the smallest impact on the norm of $w$ were then removed. Finally, the optimal number of features was found by training SVMs on subset of features, using the theoretical concept span estimate[39,40]. We used linear SVMs that were trained and tested using the R *kernlab* package[41]. For RFE, we used the function *fit.rfe* as implemented in the *pathClass* package[42]. Everything was performed in R version 3.2.3.

The application of this strategy to identify pathways that discriminate single cells receiving miR-294 or let-7c is outlined in Supplementary Fig. 5. GEPs of miRNA-transfected cells were first converted to a list of pathways (that is, features) by computing the ES of each pathway by means of a GSEA approach. Then, 1,000 different instances of the training set were randomly built by selecting five cells repeatedly from miR-294- and let-7c-transfected cells. RFE + SVM was performed for each instance of the training set, to select the most informative pathways able to discriminate the two types of cells. Finally, pathways were ranked according to the number of times they were selected by the RFE-SVMs algorithm (that is, predictive capacity). Enrichment Score (ES) and the corresponding P-value were computed by using the KS test function as in Napolitano et al.[43]

To validate the pathways identified by RFE algorithm, four different subset of discriminant pathways were generated based on different cutoffs of relevance (that is, 10, 25, 50 and 75%). Nested subsets of pathways of increasing informative density were thus obtained. The quality of these subsets of pathways was then

assessed by first training a classifier based on a linear SVM and then estimating its area under the receiver operating characteristic curve by a tenfold cross-validation model. Supplementary Fig. 5b summarizes the performance of the classifier for each combination of pathway selection. It is easy to see that the classifier preformed always better when a subset of relevant pathways is used. It is noteworthy that the best value of area under the curve is obtained when the four pathways with predicted relevance >75% are used.

**Correlation and distance analyses.** The distance among cells showed in the main Fig. 3a was quantified as 1—the absolute value of Pearson's correlation coefficient (PCC). PCC was computed using the *cor* function of the R statistical environment. Density plots were finally produced with the function *geom_desity* present in the *ggplot* package of the R statistical environment. Subpopulations of cells were identified with Dynamic tree cut package[44] in R statistical environment with default parameters and using the 'hybrid' mode with dissimilarity information among cells defined as $|1 - \text{PCC}|$.

**Cell subpopulation analysis.** The ANOVA was performed to identify differences among groups of cells within let-7c or Dgcr8 knockdown conditions. GEPs of miRNA-transfected cells were first converted to a list of pathways (that is, MsigDb hallmark gene sets) by computing the ES of each pathway by means of a GSEA approach. Thus, each gene set had a ES distribution across cells. Finally, ANOVA test among subpopulation of identified cells was performed for each gene set. Obtained *P*-values were finally corrected for multiple hypothesis testing using Benjamin and Hochberg procedure. ANOVA tets swere performed using aov function in the R statistical environment. ES and the corresponding *P*-values were computed by using the KS test function as in Napolitano *et al.*[43]

**GSEA with PCA loading values.** To find the sources of the majority of the variance among groups revealed in Fig. 1c and Supplementary Fig. 7, GSEA using MSigDb hallmark gene sets was performed ranking the genes according to their descending loadings in the principal component that shows the separation. In detail, we used PC1 loadings to explain the variance shown in Fig. 1c and Supplementary Fig. 7a, whereas we used PC3 loadings to explain the variance shown in Supplementary Fig. 7b. For each pathway the ES and the corresponding *P*-value were computed by using the KS test function as in Napolitano *et al.*[43] *P*-values were corrected for multiple hypothesis testing using Benjamini and Hochberg procedure. As positive ES corresponds to pathways containing the best discriminant genes only pathways with positive ES value and FDR <10% were considered. Significant pathways were than ranked according to their *P*-values.

**Defining miRNA targets.** The miRanda-miRSVR (August 2010 release) mouse miRNA target predictions were obtained from http://microrna.org[45]. The TargetScan[46] version 7.1 mouse miRNA target predictions were obtained from http://targetscan.org. The Melton *et al.*[5] miRNA target predictions were defined as downregulated genes in the population of cells receiving the corresponding miRNA and matching the miRNA seed complement on their the 3′-untranslated region sequences. Throughout the manuscript, unless otherwise stated, predicted miRNA targets were defined by the intersection of the downregulated genes in the population of individual cells receiving the corresponding miRNA (versus Dgcr8 knockout, adjusted *P*-value <0.1) and the ones predicted by Targetscan, miRanda or Melton *et al.*[5] (that is, union of all three; Supplementary Fig. 8 and Supplementary Data set 5). To obtain the smaller set of 'high confident' transcriptional targets, only target genes supported by at least two sources were considered. The number of miR-294 'high confident' targets was further reduced retaining only the ones downregulated (adjusted *P*-value <0.1) also in the WT versus Dgcr8-knockout conditions. This reduction in number was essential for RMI analysis.

**Estimation of RMI.** An information-theoretic approach based on RMI was used to quantify the statistical dependency (that is, co-expression) among a predefined set of genes[23]. Specifically, given a set of $d$ real-valued random variables $\mathbf{X} = (X^1, X^2, \ldots, X^d)$ with joint probability density function $f : \mathbb{R}^d \to \mathbb{R}$ and marginal densities $f_i : \mathbb{R} \to \mathbb{R}, 1 \le i \le d$, the RMI was defined for any real parameter $\alpha$, assuming the underlying integrals exist, as:

$$I_\alpha(\mathbf{X}) = I_\alpha(f) = \frac{1}{\alpha - 1} \int_{\mathbb{R}^d} \frac{f^\alpha(x^1 \ldots x^d)}{\left(\prod_{i=1}^d f_i(x^i)\right)^{\alpha - 1}} d(x^1 \ldots x^d) \quad (1)$$

when $\alpha = 1$, $I_\alpha(\mathbf{X})$ was defined in the limit $I_1 = \log_{\alpha \to 1} I_\alpha$. Indeed, the classical multi-information across $d$ variables is just a special case of RMI with $\alpha = 1$ and can be easily approximated with $\alpha = 0.99$ (ref. 47). RMI can be efficiently valued via a non-parametric estimator based on the generalized $k$ nearest-neighbour graph and copula transformation[47,48]. Briefly, that non-parametric estimator works as follow: given a collection of independent and identically distributed random variables $\mathbf{X}_{1:n} = (\mathbf{X}_1, \mathbf{X}_2, \ldots, \mathbf{X}_n)$, where each $\mathbf{X}_j = (X_j^1, X_j^2, \ldots, X_j^d)$, the algorithm

estimates the entropy $H_\alpha(f)$ for $\alpha \in (0, 1)$ as follows:

$$\hat{H}_\alpha(\mathbf{X}_{1:n}) = \frac{1}{1 - \alpha} \log \frac{L_p(\mathbf{X}_{1:n})}{\gamma n^{1 - p/d}} \text{ where } p = d(1 - \alpha) \quad (2)$$

where $L_p(\cdot)$ equals to the sum of the $p$-th power of Euclidian distance of the nodes in the nearest-neighbour graph $NN_S(\cdot)$ for some finite non-empty $S \subset \mathbb{N}^+$; $\gamma$ is a numeric constant dependent on $d, p$ and $S$ that can be estimated empirically from a large sample ($n \gg 1$). Finally, the RMI $I_\alpha$ of the $d$ variables $\mathbf{X} = (X^1, X^2, \ldots, X^d)$ from a sample of independent and identically distributed random variables $\mathbf{X}_{1:n} = (\mathbf{X}_1 \ldots \mathbf{X}_n)$ is computed as

$$\hat{I}_\alpha(\mathbf{X}_{1:n}) = -\hat{H}_\alpha(\hat{\mathbf{Z}}_1, \hat{\mathbf{Z}}_2, \ldots, \hat{\mathbf{Z}}_n) \quad (3)$$

where $\hat{H}_\alpha$ is defined as before and the sample $(\hat{\mathbf{Z}}_1, \hat{\mathbf{Z}}_2, \ldots, \hat{\mathbf{Z}}_n) = (\hat{\mathbf{F}}(\mathbf{X}_1), \hat{\mathbf{F}}(\mathbf{X}_2), \ldots, \hat{\mathbf{F}}(\mathbf{X}_n))$. $\hat{\mathbf{F}}(\cdot)$ is called empirical copula transformation[48], where the $j$-th coordinate of $\hat{Z}_i$ equals:

$$\hat{Z}_i^j = \frac{1}{n} \text{rank}(X_i^j, (X_1^j, X_2^j, \ldots, X_n^j)) \quad (4)$$

where rank$(x, A)$ is the number of elements of $A \le x$.

Further tests on the convergence of the nonparametric estimator for RMI used in this study can be found in the original paper were it was presented for the first time[47] and in our previous work[23]. Here, the above non-parametric estimator of RMI has been implemented in the R software environment version 3.2.3 and always used with the parameter $\alpha = 0.99$ and $k = 3$ (that is, number of $k$ nearest neighbours to use).

**Assessment of RMI significance.** To assess whether, in a specific population of cells, the estimated RMI value across a set $d$ miRNA targets $G^1 \ldots G^d$ was statistically significant, a permutation test corrected for the expression levels of the targets was used. Briefly, we first sorted the genes according to their average value of expression across cells and then divided the ranked list in expression bins of 500 genes. In this way, each gene is assigned to a specific bin containing genes with similar expression levels. Finally, to estimate the empirical distribution of RMI and, from that, the associated *P*-value, we randomly selected $d$ genes contained in the same bins as $G^1 \ldots G^d$ in 10,000 number of trials. For each trial, the RMI value was computed, thus obtaining its empirical distribution. The *P*-value was finally estimated as the percentage of random trials with a value of RMI greater than the one measured for the $G^1 \ldots G^d$ target genes.

**Assessment of RMI distribution among miRNA targets.** RMI estimation could not converge when the number of genes is too high respect to the number of samples available[23,47]. Thus, to show that miRNA targets tend to be co-expressed in the corresponding miRNA-transfected population, a strategy based on a bootstrapping procedure was used. Given the larger list of predicted miRNA targets (in contrast to 'high confidence' targets), we randomly extracted a subset of $d$ genes in 10,000 trials. In each trial, computed the RMI value was computed among the $d$ genes across the three populations of cells. Different value of $d$ do not affect the results (Supplementary Fig. 10).

**Differential co-expression analysis.** Changes in co-expression among a set of $d$ genes $G^1 \ldots G^d$ between two population of cells (that is, miR-294 and let-7c) was assessed by estimating the difference in RMI (or $\Delta$RMI) between the corresponding two subsets of GEPs[23] (Supplementary Fig. 11). To remove those genes whose expression was not changing significantly, a pre-filtering step was applied within each group of GEPs. Specifically, those genes whose entropy was in the fifth-percentile were excluded from the analysis. To estimate the empirical distribution of $\Delta$RMI and, from that, the associated *P*-value, we performed a permutation test. Specifically, given a gene set of $d$ genes $G^1 \ldots G^d$, the significance of the $\Delta$RMI for the gene set was computed by randomly selecting $d$ genes across 1,000 trials. For each trial, the $\Delta$RMI value was computed, thus obtaining its empirical distribution. The *P*-value was finally estimated as the percentage of random trials with a value of $\Delta$RMI greater (or lower if $\Delta$RMI < 0) than the one measured on the gene set being tested.

**Cell cycle gene-to-gene correlation.** Cell cycle gene–gene correlation was estimated by computing the PCC among 36 transcripts previously categorized to a particular cell cycle phase (Supplementary Data set 6)[49]. PCC was estimated using the *cor* function present in the R software environment 3.2.3 and plotted using *ggplot* package.

**Cell cycle stage prediction from scRNA-seq data.** Cell cycle stage of cells was predicted using the classification algorithm named 'pairs' and previously published by Scialdone *et al.*[20] This method does not require any further data normalization to be applied before and was one of the two methods performing best among the tested in the Scialdone *et al.*[20] The classifier was trained on the published single-cell RNA-seq data set comprising 182 mESCs with known cell cycle phase. Similar to our cells, the mESC of this study were cultured in LIF + 2i media[50] and thus represented a good training set to use. For the training process, the cell cycle genes used in the original manuscript were used. After the training

process, the classifier was used to assign each cell to a specific cycle stages among G1, S or G2/M from our single cell RNA sequencing data.

**Data availability.** All sequencing data can be found at GEO under the accession code GSE80168. The software code used in this study is available upon request to authors. All other data are available from the authors upon reasonable request.

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

## Acknowledgements

R.B. thanks the Telethon Institute of Genetics and Experimental Medicine (TIGEM) for hosting his sabbatical, when and where much of this work was done. We thank the TIGEM's Next Generational Sequencing Core (Dr Manuela Dionisi, Mrs Annalaura Torella and Professor Vincenzo Nigro) for library preparation and sequencing. We also thank TIGEM's Bioinformatics Core (Dr Margherita Mutarelli and Mr Veer Marwah Singh) for help mapping sequencing reads. The work was supported by funds from NIH/NIGMS (R01 GM101180) to R.B. and Telethon Foundation (TGM11SB1) to D.d.B.

## Author contributions

R.B. conceived the study. R.B. and D.d.B. directed the study. R.B. performed the experiments with help from A.C. and T.J.N. G.G. and A.C. developed the computational pipelines, G.G., A.C., L.C., D.d.B. and R.B. analysed the data. G.G., A.C., D.d.B. and R.B. wrote the manuscript.

## Additional information

**Competing financial interests:** The authors declare no competing financial interests.

**How to cite this article**: Gambardella, G. *et al.* The impact of microRNAs on transcriptional heterogeneity and gene co-expression across single embryonic stem cells. *Nat. Commun.* **8**, 14126 doi: 10.1038/ncomms14126 (2017).

**Publisher's note**: 

