## [Peer Review File · Nature Communications]

Reviewers' Comments:

Reviewer #1 (Remarks to the Author)

The manuscript "MicroRNAs shape gene co-regulation within cells and transcriptional heterogeneity across cells" by Gennaro Gambardella et al performed single cell RNA-seq for studying the effect of individual microRNA miR-294 and let-7 on gene regulatory network and transcriptional heterogeneity of mouse embryonic stem cells (ESCs). They found that both miR-194 and let-7 induced co-regulation of their target genes. They also found that these two microRNAs have opposite roles on heterogeneity of the cell population. This study well continues the authors' previously published excellent work to provide novel valuable insight into ESCC and let 7 miRNAs on ESCs gene regulatory network at the single cell transcriptome level.

Major:

- 1) Figure 1B, the PC1 well separated the cell populations. The authors could check the gene list of PC1 to get more insight into what genes separate these cell populations.
- 2) Figure 2B, the three cell subpopulations of let-7 transfected Dgcr8^{-/-} cells and Dgcr8^{-/-} cells are not well described. The authors found that the cell cycle is a main source of these subpopulations, yet they did not make clear which phase of the cell cycle each cell subpopulations is corresponding to. Also, how they determined that the subpopulations of the Dgcr8^{-/-} cells are the same as those of the let-7 transfected Dgcr8^{-/-} cells, and which subpopulation the miR-294 transfected Dgcr8^{-/-} cells are corresponding to, are not clear. Could these cell subpopulations also be identified in the PCA performed on each of these cell populations seperatedly? In addition, since let-7 has been previously shown to suppress self-renewal of ESCs, should these cell subpopulations also be related to differentiation?
- 3) The authors found that microRNAs influenced the cell cycle structure, which then affected the heterogeneity of the cell population. This is a good, but maybe not surprising finding; given that it have been already known that ESCC/let-7 microRNAs play important roles on controlling the cell cycle of ESCs and the cell cycle is known as a dominant source of cell heterogeneity. Will these microRNAs affected transcriptome heterogeneity beyond the cell cycle? The authors may use bioinformatics tools such as sLVM (PMID: 25599176), or focusing on cells of certain cell cycle phases or noncycling cells, to avoid the dominant effects of the cell cycle. In addition, should the single-cell transcriptome data provide new insight into the cell cycle control of ESCC/let7 microRNAs? For example, previous studies reported a G1/S control role of these microRNAs, while it looks that Dgcr8^{-/-} and let-7 transfected Dgcr8^{-/-} cells generated three cell subpopulations vs one major population of the miR-294 transfected cells. Could these microRNAs plays roles on aspects other than G1/S transition of the cell cycle?
- 4) Figure 3E, the authors claimed that let-7 induced and miR-294 repressed the co-regulation of cell cycle genes. Should it be possible that the simpler cell cycle structure of the miR-294 transfected cells (most cells are of just one phase of the cell cycle) leads to less variation and thus seemingly less correlation of the cell cycle genes? If so, the co-regulation of cell cycle genes is a byproduct of the changes in cell cycle structure. The authors should discuss about the possible molecular mechanism of the effects of microRNAs on the co-regulation of cell cycle genes.

Minor:

- 1) Are the concentrations of the transfected microRNAs miR-294 and let-7 comparable with their physical concentration? If transfected concentration is too much higher than the physical concentration, artificial effects should be concerned. The authors could examine this by using northern blotting or other methods.

Reviewer #2 (Remarks to the Author)

Using single-cell gene expression profiles, the authors describe evidence that transfections of miR-

294 and Let7C into miRNA-depleted mESCs alter intra-cellular heterogeneity. The experiments are not necessarily physiologically informative because of the extreme activation of these miRNAs due to low competition for RISC, but they demonstrate extreme effects that may be due by regulation of these miRNAs. The authors attempt to show that the effects are enriched for natural targets of these miRNAs, although their evidence is inconclusive. The novelty of the study is the study of intra-cellular heterogeneity, and the most convincing evidence is given in Figure 2A, but its presentation is incomplete (see below). The authors also conclusively validate predicted miR-294 and Let7C, on the population level.

Without more detailed tests regarding the statistics behind the figures and detailed more discussion of the interpretation of the results, I don't see how evidence provided conclusively supports the author's conclusions. If my conclusion is wrong, it will indicate that the clarity of the arguments can be improved. I do believe that the experiments are very interesting and would like to see the authors improve their manuscript, which does have the potential to be an asset to the miRNA community, and has some components that are carefully and well exposed. Since the authors focused the presentation around 3 figures, I organized all major critiques by figure, below.

Figure 1: Figure 1C shows differential regulation of genes after transfecting miRNAs and miRNA targets are shown as triangles. miRNA targets are not identified in Table S1, and appear to be down regulated at 100%. This seems unlikely. Where there are no upregulated predicted miRNA targets? It's OK if there were some, but I expect to have that information given that the authors chose to make this argument. In addition, in the plot of DGCR8^{-/-} vs. WT the caption says that only miR-294 targets are shown. If that so, why? There is no indication that of this in the figure itself. I would at least expect to see both let7c and miR-294; possibly all genes with 3' UTRs, or no binding targets at all.

Figure 2: Figure 2A shows that miR-294 increases correlation between cells relative to Mock transfected controls. Can you produce a p value? What was mock? Is it possible that mock is sporadically targeting genes? Let7c transfections appear to be statistically indistinguishable from controls for all genes but increase correlation between lineage regulators. The authors state that it decreases correlation, but I see no evidence for this from the text or the plot. Figure 2B is very hard to understand; is let7c really producing more distinct populations than mock?

Figure 3: Doesn't evidence shown in this figure simply demonstrate regulation of the miRNAs and enrichment of their targets in specific pathways? Based on this evidence wouldn't any regulator "establish de novo networks of co-regulated genes"?

Reviewer #3 (Remarks to the Author)

A. Summary of the key results

This paper shows how miRNAs affect co-regulation of genes within cells, and the transcriptional heterogeneity across cells, by single-cell transcriptome sequencing after introducing miRNAs in double-knockout Dgcr8 cells (which due to the absence of Dgcr8 do not contain endogenous miRNAs). As expected, the miRNAs suppressed their target genes, but also affected coregulation of their target genes. The two miRNAs tried had different effects, with one increasing heterogeneity and the other decreasing heterogeneity. The authors suggest that this is due to one miRNA promoting and the other repressing cell cycle phasing.

B. Originality and interest: if not novel, please give references

Coregulation of genes targeted by the same miRNA has been demonstrated previously, as in references 9-12 cited by the authors. The novelty is that this is demonstrated here in single cells. For miR-294, the single-cell results show an decrease in heterogeneity, which would also be

expected in bulk studies. It is a nice confirmation, but it is not surprising. On the other hand, let-7c increases the heterogeneity. The authors cite reference (15) to suggest that this is not due to differentiation, but fail to test this experimentally. Instead, the authors find that let-7c affects the phasing of the cell cycle. This conclusion is independent of the concept of coregulation of genes by miRNAs. While these two case studies are interesting by themselves, I don't see what the general conclusions I can learn from this manuscript, or how the two cases are related to each other.

C. Data & methodology: validity of approach, quality of data, quality of presentation

The text in many places is difficult to follow.

In lines 45-56: Why is widespread targeting more prominent among large miRNA families? If they have the same seed sequence, don't the miRNAs in the same family have the same targets? Or do miRNAs occurring in large families have more targets?

In lines 51-52: On first reading, it is not clear what is meant by miRNAs linking genes into networks.

In line 57: single cell sequencing of what?

Lines 61-65: The technical details can be skipped in the main text.

In line 72-73: I agree that the function of miR-294 was confirmed, but how was the function of let-7c confirmed?

In line 90: "principle" should be "principal".

In line 93: in the differentially expressed genes, do you include both genes that were upregulated and genes that were downregulated?

In line 102: add a comma between "uncovered" and "including". Also, note the typo in mesenchymal.

In line 129: At this point in the text, readers will not understand what is meant by a structured cell cycle.

In line 192: What are "2i culture conditions"?

In line 315: What is the RFE+SVMs method?

D. Appropriate use of statistics and treatment of uncertainties

P-values are not always shown explicitly, for example in lines 111-117 and in line 148.

In line 167-168: Is the difference between let-7c and mock-transfected cells significant?

In lines 132-147: To what extent do the conclusions depend on using the RMI instead of simply the average correlation?

E. Conclusions: robustness, validity, reliability

I am not convinced of the conclusions.

Cell cycle is often enriched in perturbation experiments just due to the stress imposed on the cell. Indeed, the mock-transfected cells also show a separation by cell cycle phase (Figure 3E central panel). So I am not convinced that this is due to let-7c. But if it is, then what is the role of coregulation of genes for this effect? Isn't it just due to let-7c downregulating its target genes?

F. Suggested improvements: experiments, data for possible revision

In line 192-193: Can you demonstrate in your experiment that let-7c promotes ESC self-renewal?

G. References: appropriate credit to previous work?

Yes.

H. Clarity and context: lucidity of abstract/summary, appropriateness of abstract, introduction and

conclusions

In the title, it is not clear what "microRNAs shape gene co-regulation" means. Overall, in this manuscript the definition of co-regulation is unclear, though this is the central theme. I don't know now if this means that one miRNA regulates many target genes, or that target genes regulate each other because each one acts as a microRNA sponge for the other target genes. This term should be clearly defined in the abstract and in the main text. The conclusions need to be strengthened to show the overall message of the paper, which I feel currently is lacking.

Point by Point reply to Reviewers' comments.

Reviewer #1

The manuscript "MicroRNAs shape gene co-regulation within cells and transcriptional heterogeneity across cells" by Gennaro Gambardella et al performed single cell RNA-seq for studying the effect of individual microRNA miR-294 and let-7 on gene regulatory network and transcriptional heterogeneity of mouse embryonic stem cells (ESCs). They found that both miR-194 and let-7 induced co-regulation of their target genes. They also found that these two microRNAs have opposite roles on heterogeneity of the cell population. This study well continues the authors' previously published excellent work to provide novel valuable insight into ESCC and let 7 miRNAs on ESCs gene regulatory network at the single cell transcriptome level.

Major:

1.1) Figure 1B, the PC1 well separated the cell populations. The authors could check the gene list of PC1 to get more insight into what genes separate these cell populations.

Response: We thank the reviewer for the useful suggestion. In order to get more insight into which biological process separate these cell populations, we have now performed a Gene Set Enrichment Analysis (GSEA) using the PC1 loading values. GSEA revealed a strong enrichment for cell cycle related genes. GSEA was performed using the hallmarks gene-sets annotated in the MsigDb database that describe genes with coordinated expression levels in a variety of tissues involved in specific biological processes (1). We have now added a new Supplementary Table S2 to report the GSEA analysis and added text in the Result section (page 4 lines 140-144) and in the Methods section (page 12, paragraph *PCA analysis and differential expression*).

1.2) Figure 2B, the three cell subpopulations of let-7 transfected and Dgcr8^{-/-} cells are not well described. The authors found that the cell cycle is a main source of these subpopulations, yet they did not make clear which phase of the cell cycle each cell subpopulations is corresponding to.

Response: As suggested by the reviewer, we have now performed additional analyses to determine the cell cycle phase for each of the cells in our experiments. Specifically, we analysed single-cell data with Cyclone, a recently published software by Scialdone et al. 2015 (2), able to determine the cell cycle phase from single-cell transcriptome data (Material and Methods, page 13-14 new paragraph *Cell subpopulation analysis*). Cyclone uses a machine-learning method and a set of transcriptional biomarkers to predict the cell cycle phase with a good level of accuracy in ES cells. We found a significant reduction of the G1 phase in miR-294 cells and a significant expansion of the G1 phase in let-7c cells, when compared with Dgcr8^{-/-} cells (revised Fig. 3e). This result is in agreement with our previous findings (3, 4). We then applied Cyclone to each of the three subpopulations of let7 and Dgcr8^{-/-} cells. Upper panel of revised Fig. 3b reports the percentage of cells in each of the cell cycle phases for the subpopulation of cells in let7-c and Dgcr8^{-/-}. In let-7c cells, subpopulation alpha consists of cells mostly in the G1 phase (about 80%); subpopulation gamma is composed only by cells in G1/S phase, while subpopulation beta is the most heterogeneous and composed by cells in all the three phases. Cell-cycle phase heterogeneity is present in all of the three Dgcr8^{-/-} subpopulations. Hence, we conclude that while in let-7c cells the three subpopulations do differ in the cell cycle phase, the same is not true for Dgcr8^{-/-} subpopulations, where cell cycle phase is not the main source of variation. We have modified the Results and Discussion sections to illustrate these new results.

1.3) Also, how they determined that the subpopulations of the Dgcr8^{-/-} cells are the same as those of the let-7 transfected Dgcr8^{-/-} cells, and which subpopulation the miR-294 transfected Dgcr8^{-/-} cells are corresponding to, are not clear.

Response: We apologize for causing confusion. In the first version of this manuscript, we used the same symbols when referring to both the subpopulations of let-7c cells and Dgcr8^{-/-} cells, thus implying that these subpopulations were somehow related. We have now modified Figure 3b and its caption to make clearer that the three subpopulations of cells identified in Dgcr8^{-/-} cells are not at all related to the three subpopulations of let-7c cells.

1.4) Could these cell subpopulations also be identified in the PCA performed on each of these cell populations separately?

Response: We have now performed a PCA as reported in the Supplementary Fig. 7. Let-7c transfected Dgcr8^{-/-} cells shows a clear separation on PC1, while for Dgcr8^{-/-} cells a separation of the three cell subpopulations is present on PC3. Gene Set Enrichment Analysis (GSEA) using PC1 loading values of let-7c transfected cells revealed again a strong enrichment for cell cycle related genes emphasizing that cell cycle genes are major drivers in the appearance of subpopulations in let-7c transfected cells (new Supplementary Table 2). GSEA for Dgcr8^{-/-} cells using PC3 loading values were instead related to translation and RNA processing (new Supplementary Table 2).

1.5) In addition, since let-7 has been previously shown to suppress self-renewal of ESCs, should these cell subpopulations also be related to differentiation?

Response: To address this point, we checked the expression of 16 genes known to be markers of either differentiation (Sox17, Brachyury, Fgf5, Sox1, Pax6, Grhl2, Mixl1, Gata4, Gata6, Foxa1) or the pluripotency state (Pou5f1, Nanog, Sall4, Esrrb, Klf4 and Rex1) in mESC. Differentiation markers remain repressed and the pluripotency markers remain highly expressed in the let-7c transfected cells (revised Figure 1d). The same observation holds true also when analysing the markers for the three individual let-7c subpopulations (revised Figure 3f). While we have previously shown that let-7c induces differentiation (5), this result was obtained when growing mES cells in the absence of 2i, conversely to the present work where cells were grown in presence of 2i. Furthermore, Collins, Daley, and colleagues (6) recently showed that let-7 does not negatively impact colony reformation, in the setting of 2i, consistent with our results. In summary, 2i appears to override let-7's ability to induce differentiation and, therefore the changes we see here cannot be attributed to differentiation. We include the marker expression in new Figure 1d and Figure 3f.

1.6) The authors found that microRNAs influenced the cell cycle structure, which then affected the heterogeneity of the cell population. This is a good, but maybe not surprising finding; given that it has been already known that ESCC/let-7 microRNAs play important roles on controlling the cell cycle of ESCs and the cell cycle is known as a dominant source of cell heterogeneity. Will these microRNAs affect transcriptome heterogeneity beyond the cell cycle? The authors may use bioinformatics tools such as scLVM (PMID: 25599176), or focusing on cells of certain cell cycle phases or noncycling cells, to avoid the dominant effects of the cell cycle.

Response: Yes, we have previously shown that let7c increases while miR-294 decreases number of cells in G1 as defined by DNA content (3, 4). As mentioned above in reply to point 1.2, we now include new analysis using the Cyclone software, which confirms the change in distribution of cells across the cell cycle with the addition of the miRNAs (revised Figure 3e). However, in this paper we take this finding significantly further by showing that not only do we change the proportion of cells in each phase of cell cycle, but we also change the co-expression of cell cycle phase genes. This increase in co-expression cannot be solely ascribed to change in the distribution of cells across cell cycle. We have changed the visualization of the data in the revised Fig. 4d-e to make this argument clearer. For example, as shown in revised Fig. 3e, let-7c reduces the number of cells in G2/M, but Fig. 4d-e shows that it increases the co-expression of G2/M genes. The converse is seen with miR-294 (increased cells in G2/M, but reduced co-expression of G2/M

genes). Thus, co-expression of these genes cannot be simply ascribed to an increase in number of cells in that phase. We have now modified the main text and revised Figure 3 and 4 in the Results and Discussion sections to make this point clearer.

As for using scLVM to look for cell cycle independent co-regulated genes, we have tried it. However, we could not make much sense of the data, and recent work by McDavid et al has shown that the interpretation of this analysis is complicated by the fact that most variation uncovered by scLVM can be explained by geometric library size rather than cell cycle stage (McDavid, A., Finak, G. & Gottardo, R. *Nat. Biotechnol* **34**, 595–597 (2016)). Given the confusing output and controversy around the approach, we have chosen not to pursue this course.

1.7) In addition, should the single-cell transcriptome data provide new insight into the cell cycle control of ESCC/let7 microRNAs? For example, previous studies reported a G1/S control role of these microRNAs, while it looks that Dgcr8^{-/-} and let-7 transfected Dgcr8^{-/-} cells generated three cell subpopulations vs one major population of the miR-294 transfected cells. Could these microRNAs play roles on aspects other than G1/S transition of the cell cycle?

Response: Yes, we believe these miRNAs do play roles on aspects other than the G1/S transition. We show in revised Fig. 4f that the miRNAs induce the co-expression of genes involved in multiple pathways. Indeed, miR-294 promotes co-expression of genes in 12 ontogeny groups (revised Fig. 4f). One of these pathways has been previously shown to be an important target of the ESCC family (the epithelial-mesenchymal transition) (7, 8), providing additional confirmation of how the miRNA induced co-expression is associated with cellular outcome. The other pathways are novel targets of the miRNAs. In this paper, we focus on cell cycle because of the striking impact of the miRNAs. Furthermore, while the concept of diminished co-expression of cell cycle phase genes in ESCs relative to somatic cells has been described (9), the remarkable impact of miRNAs on the co-expression had not. Therefore, it was a natural story to follow.

1.8) Figure 3E, the authors claimed that let-7 induced and miR-294 repressed the co-regulation of cell cycle genes. Should it be possible that the simpler cell cycle structure of the miR-294 transfected cells (most cells are of just one phase of the cell cycle) leads to less variation and thus seemingly less correlation of the cell cycle genes?

Response: As mentioned in to response to point 1.2, we have added Cyclone analysis of the data to determine the proportion of cells in each phase of cell cycle. Let-7c increases number of cells in G1, but decreases number of cells in G2 (revised Figure 3e). In contrast, miR-294 decreases number of cells in G1, but increases number of cells in G2 (revised Figure 3e). Therefore, it is not simply a matter of miR-294 cells having a simpler cell cycle. Importantly, as mentioned in our reply to point 1.6, the co-expression of cell cycle phase genes does not simply follow distribution of cells across cell cycle. Let-7c induces co-expression of the genes in all phases of the cell cycle, while miR-294 does that opposite (revised Figure 4d-e).

1.9) If so, the co-regulation of cell cycle genes is a by-product of the changes in cell cycle structure. The authors should discuss about the possible molecular mechanism of the effects of microRNAs on the co-regulation of cell cycle genes.

Response: We agree that understanding the mechanism for let-7c increasing and miR-294 reducing the co-expression of genes independent of cell cycle structure is a very exciting avenue of research. We do know that the miRNAs directly target some cell cycle phase genes, but how that translates to a larger network of co-expressed cell cycle phase genes is unclear and certainly worthy of further study, but we believe beyond the scope of the paper. We believe that using single cell sequencing to show that miRNAs influence the co-expression of genes in pathways that are part of the biology of these cells is already an important advance.

Minor:

1.10) Are the concentrations of the transfected microRNAs miR-294 and let-7 comparable with their physical concentration? If transfected concentration is too much higher than the physical concentration, artificial effects should be concerned. The authors could examine this by using northern blotting or other methods.

Response: Yes, we believe they are comparable. The ESCC seed sequence normally makes up 60-70% of the total miRNA population in ESCs. The amount of active ESCC is limited by the availability of Ago and, if anything, Ago is down in the Dgcr8 knockout cells. Most importantly, expression analysis of the single cells shows that the miR-294 transfected cells almost perfectly overlap with wild type cells (revised Fig. 1c). Indeed, they are basically indistinguishable. These facts strongly support that the transfected miR-294 is acting at near physiological levels. As for let-7c, this miRNA is not normally expressed in ESCs, so obviously the transfected levels do not represent physiological levels in that context. However, in somatic cells where let-7c is highly expressed, it typically makes a large portion of the miRNA population (eg. 40% in mouse embryonic fibroblasts) and thus in that sense the transfected miRNA represents near physiological levels. We now discuss this issue in the text (paragraph *Single cell sequencing of cell* at page 3 and 4).

Reviewer #2:

2.1) Using single-cell gene expression profiles, the authors describe evidence that transfections of miR-294 and Let7C into miRNA-depleted mESCs alter intra-cellular heterogeneity. The experiments are not necessarily physiologically informative because of the extreme activation of these miRNAs due to low competition for RISC, but they demonstrate extreme effects that may be due by regulation of these miRNAs.

Response: This issue was addressed in our response to Reviewer 1 (point 1.10). In brief, the ESCC miRNAs make up upwards of 70% of the miRNAs in ESCs. As miRNA function is limited by Ago concentration, which is down in miRNA-deficient mutants (10), it is hard to imagine that we would be able to produce supraphysiological doses of this miRNA family. Phil Sharp and colleagues recently published that the ESCC and let-7c miRNAs are already at saturating doses in terms of their targets in their respective cell types (11). Therefore, even elevated levels of the miRNAs are unlikely to have any additional effect on downstream targets. Indeed, our single cell expression analysis shows that the introduction of miR-294 into Dgcr8 knockout cells makes them indistinguishable from wild type cells supporting physiological function of the introduced miRNA (revised Figure 1b-c).

The authors attempt to show that the effects are enriched for natural targets of these miRNAs, although their evidence is inconclusive.

Response: We are not sure what evidences the Reviewer is referring to. In previous Fig. 1c (now revised Fig. 2a), we show that the known targets of miR-294 and let7c are indeed significantly downregulated in the scRNA-seq data in their respective transfection experiments when compared to untransfected Dgcr8^{-/-} cells. In the revised Fig. 1b, we now report the result of Gene Set Enrichment Analysis (GSEA) (12) that we have performed for each individual cell to further demonstrate that the targets of mir-294 and let-7c are indeed repressed in each cell as expected in their respective experiments. The GSEA outputs an Enrichment Score that quantifies how much a set of genes tends to be coherently up- (positive scores) or down-regulated (negative score). It can be appreciated in Fig. 1b that miR-294 targets are consistently downregulated in the miR-294 transfection experiment and in the WT cells, in agreement with the fact that miR-294 is highly expressed in wild type stem cells. On the contrary, let-7c targets are down-regulated only in the let7c experiments. These results clearly show that the transcriptional changes we measure are indeed caused by the microRNAs we transfected.

2.2) The novelty of the study is the study of intra-cellular heterogeneity, and the most convincing evidence is given in Figure 2A, but it the presentation is incomplete (see below). The authors also conclusively validate predicted miR-294 and Let7C, on the population level. Without more detailed tests regarding the statistics behind the figures and detailed more discussion of the interpretation of the results, I don't see how evidence provided conclusively supports the author's conclusions. If my conclusion is wrong, it will indicate that the clarity of the arguments can be improved.

Response: In our revised manuscript, we have added additional statistical values and have provided more detailed discussion of the interpretation of results to improve clarity as outlined to reviewer's specific comments below.

I do believe that the experiments are very interesting and would like to see the authors improve their manuscript, which does have the potential to be an asset to the miRNA community, and has some components that are carefully and well exposed. Since the authors focused the presentation around 3 figures, I organized all major critiques by figure, below.

2.3) Figure 1: Figure 1C shows differential regulation of genes after transfecting miRNAs and miRNA targets are shown as triangles. miRNA targets are not identified in Table S1, and appear to be down regulated at 100%. This seems unlikely. Where there no upregulated predicted miRNA targets? It's OK if there were some, but I expect to have that information given that the authors chose to make this argument.

Response: We apologize for not having been clearer. In the previous Figure 1c1c (now revised Fig. 2a), we highlighted only statistically significant differentially expressed targets, i.e. with a FDR<10% (not all the targets). Indeed, not all the targets were significantly differentially expressed, but those that were, were indeed downregulated. This finding is not that surprising given the fact that the targets we used for our analysis, for both miR-294 and let-7c, were taken from our previous study (5) where we defined as targets those bearing a seed in their 3'UTR and whose expression decreased upon miRNA transfection. We have now revised Figure 1c to include all targets (both significant and not-significant) and identified the targets in Supplementary Table 1.

2.4) *In addition, in the plot of DGCR8^{-/-} vs. WT the caption says that only miR-294 targets are shown. If that so, why? There is no indication that of this in the figure itself. I would at least expect to see both let7c and miR-294; possibly all genes with 3' UTRs, or no binding targets at all.*

Response: We have now added in the revised Figure 2a a volcano plot including also the let-7c targets in the Dgcr8^{-/-} vs wild type experiment. As expected let-7c targets are largely unchanged between Dgcr8^{-/-} and wild type (wt) ESCs. The reason is that let-7c is not expressed in wild type ESCs. In contrast, the ESCC miRNAs (represented by miR-294 here) are the predominant miRNA in ESCs and, therefore, as expected its targets are up in the knockout.

2.5) *Figure 2: Figure 2A shows that miR-294 increases correlation between cells relative to Mock transfected controls. Can you produce a p value?*

Response: The previous Figure 2a has been now revised and it is now Figure 3a. To show that miR-294 and let-7c are respectively reducing and increasing variability across cells compared to miRNA deficient cells Dgcr8^{-/-} we performed a one-tailed Mann-Whitney test for each comparison. Specifically, the Mann-Whitney test shows that the addition of miR-294 significantly increased the correlation between cells (P<2.2e-16) relative to the miRNA deficient cells Dgcr8^{-/-}. On the contrary, let-7c decreased the correlation between cells (P<2.2e-16). These tests have been added in the updated version of the main text and in Figure 3a (inset in the upper panel).

2.6) *What was mock?*

Response: We apologize for causing confusion by improperly using the term “mock” in the previous Figure 2 and elsewhere in the text. *Mock* was used to indicate untransfected Dgcr8^{-/-} cells. We have now corrected the Figure and the text replacing *Mock* with Dgcr8^{-/-}.

2.7) *Is it possible that mock is sporadically targeting genes?*

Response: As we wrote in the point above, *Mock* was our internal name to indicate untransfected Dgcr8^{-/-} cells. Again we apologize for the confusion.

2.8) *Let7c transfections appear to be statistically indistinguishable from controls for all genes but increase correlation between lineage regulators. The authors state that it decreases correlation, but I see no evidence for this from the text or the plot.*

We appreciate that it may not be immediately obvious in previous Figure 2A (now Figure 3a) that let-7c is decreasing correlation among cells relative to Dgcr8^{-/-} cells. However, the peak of the let7c distribution is clearly shifted to the left of the peak of the Dgcr8^{-/-} distribution. We now added an inset in Figure 3a upper panel with significance values to show that this shift is indeed highly significant. As for the lineage regulators, we took that list from the recent manuscript of Collins & Daley for consistency sake (6).

However, since lineage regulators were not expressed in these cells, it is sort of meaningless and, therefore, we have now removed it in the revised manuscript.

2.9) Figure 2B is very hard to understand; is let7c really producing more distinct populations than mock?

Response: Previous Figure 2B has been revised and it is now Figure 3b. Indeed, the demarcation between cell subpopulations is much stronger in let-7c cells than in Dgcr8^{-/-} cell. In order to demonstrate this observation, we have now added a quantitative analysis of the distances between the subpopulations in the let-7c transfected cells and the Dgcr8^{-/-} cells (revised Fig. 3c). Briefly, we computed the inter-cluster distance between clusters obtained with the hierarchical clustering algorithm that we used to identify cell subpopulations (Fig. 3b). Inter-cluster distance for a cell x is defined as its average distance from all the other cells, except the ones in the same subpopulation as cell x . The distance between two cells is defined as $1 - |\text{corr}(x,y)|$. Figure 3c shows that the intra-cluster distance of let-7c cells is significantly higher than the intra-cluster distance of Dgcr8^{-/-} cells (two tailed Mann-Whitney test $P=1.94e-8$), implying that let-7c is really producing more distinct and better separated sub-populations of cell compared to the ones identified in Dgcr8^{-/-} miRNA deficient cells.

2.10) Figure 3: Doesn't evidence shown in this figure simply demonstrate regulation of the miRNAs and enrichment of their targets is specific pathways? Based on this evidence wouldn't any regulator "establish de novo networks of co-regulated genes"?

Response: Previous Figure 3 is now Figure 4. We believe part of the confusion here was caused by our improper use of the words "co-regulation" and "co-expression". By co-expression among a set of genes, we mean that the expression of the genes in the set is correlated across cells; that is, the level of expression of one gene is predictive of the level of expression of the other genes in the same cell. The term *co-regulation* implies that the co-expression among genes is caused by the presence of a common regulator, therefore is less generic and more specific.

In regards to previous Figure 3a-c (Figure 4a-c in the revised manuscript), where we focus on the miRNA targets, we are showing that introduction of the miRNA is inducing co-expression of their targets (i.e. co-regulation). To our knowledge we are the first to show that this is the case in single-cell data. The observation that microRNA targets become co-expressed across single cells in the presence of their cognate microRNA might be expected but the reason why this happens is not trivial. It is unlikely due to limiting amounts of the miRNA as both exogenously introduced levels in Dgcr8^{-/-} cells and endogenous levels in wild type ESCs (for miR-294) are very high. One possible explanation is that there is some rate-limiting factor in each cell that determines how much the targets can be degraded (eg. Varying levels of Ago from one cell to the next). This can lead to the development of 'waiting lines' for biochemical processing which may cause the appearance of correlations. This hypothesis has been put forward by Hasty and colleagues at UCSD to explain correlation between two unrelated proteins (not transcripts) tagged for degradation by the proteasome (13). We believe that maybe a similar or related mechanism is occurring here and is discussed in the discussion.

In terms of previous Figure 3d (now Figure 4f in the revised manuscript), where we evaluate co-expression of the hallmark gene-sets by RMI, each of these gene-sets has few if any direct targets (new Supplementary Fig. 13). Moreover, the effects are present even after removing the known targets of each miRNA from tested gene sets (Supplementary Fig. 12). Meaning that, probably the co-expression of the genes in these pathways is the result of indirect effects rather than direct effects on the targets. The nature of these indirect effects remains to be determined as discussed in response to reviewer 1, point 1.9.

Reviewer #3:

3.1) A. Summary of the key results

This paper shows how miRNAs affect co-regulation of genes within cells, and the transcriptional heterogeneity across cells, by single-cell transcriptome sequencing after introducing miRNAs in double-knockout Dgcr8 cells (which due to the absence of Dgcr8 do not contain endogenous miRNAs). As expected, the miRNAs suppressed their target genes, but also affected coregulation of their target genes. The two miRNAs tried had different effects, with one increasing heterogeneity and the other decreasing heterogeneity. The authors suggest that this is due to one miRNA promoting and the other repressing cell cycle phasing.

3.2) B. Originality and interest: if not novel, please give references

Co-regulation of genes targeted by the same miRNA has been demonstrated previously, as in references 9-12 cited by the authors. The novelty is that this is demonstrated here in single cells. For miR-294, the single-cell results show a decrease in heterogeneity, which would also be expected in bulk studies. It is a nice confirmation, but it is not surprising.

Response: We are unclear why it would be expected from bulk studies. Indeed, bulk studies do not allow one to assess heterogeneity in gene expression across cells, as cells' mRNA are sequenced together in bulk. We are unaware of previous work showing that a particular miRNA decreases cell-to-cell heterogeneity, especially not at a genome-wide level. Please also read our reply to Reviewer 2, point 2.10.

3.3) On the other hand, let-7c increases the heterogeneity. The authors cite reference (15) to suggest that this is not due to differentiation, but fail to test this experimentally.

Response: We now include additional data (revised Fig. 1d and Figure 3f) showing that the cells do not differentiate with introduction of let-7. Please read also our response to Reviewer 1, point 1.5.

3.4) Instead, the authors find that let-7c affects the phasing of the cell cycle. This conclusion is independent of the concept of co-regulation of genes by miRNAs. While these two case studies are interesting by themselves, I don't see what the general conclusions I can learn from this manuscript, or how the two cases are related to each other.

Response: In this manuscript, the main question we wanted to investigate was if and how microRNAs affect cell heterogeneity and gene co-expression across single cells. The transcriptomic effects of introducing a single microRNA in microRNA deficient Dgcr8^{-/-} cells have never been investigated at the single cell level. Our main conclusions are that: (i) cells can become more similar (miR-294) or more heterogeneous (let-7c) depending on the microRNA and the cellular context (revised Fig. 2); (ii) the main source of heterogeneity is cell-cycle phases (revised Fig. 3); (iii) miRNAs induce co-expression of their target genes (revised Fig. 4). (iv) miRNAs induce co-expression of genes in pathways related to the biology of cell. These genes are not necessarily direct targets of the miRNAs (new Supplementary Fig. 12) and, therefore, are indirect consequences of miRNA targeting (new Supplementary Fig. 13). The nature of these indirect effects remains to be determined as discussed in response to reviewer 1, point 1.9.

Please read also our replies to Reviewer 1 to point 1.6, 1.7, 1.8 and 1.9. We have made many changes to the text to help clarify these points.

3.5) C. Data & methodology: validity of approach, quality of data, quality of presentation

The text in many places is difficult to follow. In lines 45-56: Why is widespread targeting more prominent among large miRNA families? If they have the same seed sequence, don't the miRNAs in the same family have the same targets? Or do miRNAs occurring in large families have more targets?

Response: Sorry for the confusion. It is that miRNAs occurring in large families tend to have more targets. This point has been removed as it is not necessary for the arguments of this paper.

3.6) In lines 51-52: On first reading, it is not clear what is meant by miRNAs linking genes into networks.

Response: We agree that this might be a confusing phrase. What we mean by the phrase is that the miRNAs are inducing co-expression of sets of genes. By co-expression we mean that the expression of the genes become correlated across cells, that is the level of expression of one gene is predictive of the level of expression of the other genes in the same cell. We have now clarified this concept in the text and avoid the phrases *linking genes into networks* or *co-regulation*, which the reviewers found confusing.

3.7) In line 57: single cell sequencing of what?

Lines 61-65: The technical details can be skipped in the main text.

In line 90: "principle" should be "principal".

In line 102: add a comma between "uncovered" and "including". Also, note the typo in mesenchymal.

In line 192: What are "2i culture conditions"?

In line 315: What is the RFE+SVMs method?

Response: We have now addressed all these suggested changes in revised text.

3.8) In line 129: At this point in the text, readers will not understand what is meant by a structured cell cycle.

Response: We have reworded the sentence to make it clearer.

3.9) In line 72-73: I agree that the function of miR-294 was confirmed, but how was the function of let-7c confirmed?

Response: We agree and reworded accordingly. We were just trying to convey that based on labelled mimic FACS and changes in cell size with miR-294 that our miRNAs were getting in the cell and have an impact on the cells. I guess at this stage of the paper, one could argue that let-7c could behave differently and for some strange reason not get in the cell and/or not function in the cell. However, later in the paper, we present data that clearly shows that is not the case based on: (i) the confirmation that the let-7c repressed its own targets (revised Figure 1b); (ii) it increase the co-expression of its targets (revised Figure 4a-c) and (iii) it affects the proportion of cells in each cell cycle phases (revised Figure 3e) as we previously demonstrated (3, 4).

3.10) In line 93: in the differentially expressed genes, do you include both genes that were upregulated and genes that were downregulated?

Response: As described in our response to Reviewer 2 for the point 2.3, we had previously *only* highlighted the significantly differentially expressed miRNA targets (FDR <10%). We now highlight all the targets in the volcano plots (revised Figure 2a). These are targets that were previously found repressed in bulk cell population studies (Supplementary Table 1) and thus we would not expect any to be significantly upregulated. The fact that not all these targets are found to significantly downregulated likely reflects the reduced sensitivity of single cell sequencing when comparing differential expression across conditions.

3.11) D. Appropriate use of statistics and treatment of uncertainties

P-values are not always shown explicitly, for example in lines 111-117 and in line 148.

Response lines 111-117: We have now performed additional statistical analyses to explicitly report p-values where appropriate. In the specific example mentioned by the reviewer, as we wrote in our reply to Reviewer 2, point 2.5, in order to show that miR-294 and let-7c are respectively reducing and increasing variability across cells compared to miRNA deficient cells *Dgcr8*^{-/-} we performed a one-tailed Mann-Whitney test for each comparison. Specifically, the Mann-Whitney test shows that the addition of miR-294 significantly increased the correlation between cells ($P < 2.2 \times 10^{-16}$) relative to the miRNA deficient cells *Dgcr8*^{-/-}. On the contrary, let-7c decreased the correlation between cells ($P < 2.2 \times 10^{-16}$). These tests have been added in the updated version of the main text and in the revised Figure 3a (inset in upper panel).

Response lines 148: We have now reported explicitly in the main text the p-values that was previously only reported in the previous Figure 3a-b (Figure 4a-b in the revised manuscript).

3.12) *In line 167-168: Is the difference between let-7c and mock-transfected cells significant?*

Response: Yes, and we now add an additional panel to revised Figure 4 in order to show this effect more clearly. Specifically, in the revised Figure 4e we show how the co-expression of cell cycle phase-specific genes is affected by let-7c and miR-294 and indicated with an asterisk the significant differences between let-7c against *Dgcr8*^{-/-}, and miR-294 against *Dgcr8*^{-/-}. Again, we apologize for the use of the word *mock* in the original version of the manuscript. Mock actually refers to *Dgcr8*^{-/-} cells, which have not been transfected with any miRNA.

3.13) *In lines 132-147: To what extent do the conclusions depend on using the RMI instead of simply the average correlation?*

Response: In order to answer this question, we measured in each of the three conditions (miR-294, let-7c or *Dgcr8*^{-/-}) the Pearson Correlation Coefficient (PCC) between each pair of target genes both for miR-294 targets and let-7c targets and then computed the average correlation for each set of targets across the three conditions. The results are shown in the new Supplementary Fig. 9. It can be appreciated that whereas an effect can still be detected when comparing miR-294 targets and let-7c targets within each condition (Supplementary Fig. 9a), that is miR-294 targets are on average more correlated than let-7c targets in miR-294 transfected cells, and vice-versa in let-7c transfected cells (Supplementary Fig. 9a), the same is not true across conditions (Supplementary Fig. 9b). That is, miR-294 targets do not appear to be significantly more correlated on average in the miR-294 transfected cells when compared to let-7c transfected cells or *Dgcr8*^{-/-} cells. The same observation was present for let-7c cells. These results show that RMI is a more robust estimator of co-expression than the average pairwise correlation among targets (Supplementary Fig. 9c-d).

3.14) E. Conclusions: robustness, validity, reliability

I am not convinced of the conclusions. Cell cycle is often enriched in perturbation experiments just due to the stress imposed on the cell. Indeed, the mock-transfected cells also show a separation by cell cycle phase

(Figure 3E central panel). So I am not convinced that this is due to let-7c. But if it is, then what is the role of coregulation of genes for this effect? Isn't it just due to let-7c downregulating its target genes?

Response: We apologize for causing confusion by improperly using the term “mock” in the Figure 2 and elsewhere in the text. *Mock* was used to indicate untransfected Dgcr8^{-/-} cells and not mock-transfected cells, which are not present in our experiments.

As discussed in our response to Reviewer 1, point 1.6 and point 1.8, we have previously shown that let-7c increases while miR-294 decreases number of cells in G1 as defined by DNA content (3, 4). We now include new analyses using the Cyclone software, which confirms the change in distribution of cells across the cell cycle with the addition of the miRNAs (revised Figure 3e). However, in this paper we take this finding significantly further by showing that not only do we change the number of cells in each phase of cell cycle, but we also change the co-expression of cell cycle phase genes (revised Figure 4d-e). This increase in co-expression cannot be solely ascribed to change in the distribution of cells across cell cycle. We have changed the visualization of the data in the revised Fig. 4d-e to make this argument clearer. For example, as shown in revised Fig. 3e, let-7c reduces the number of cells in G2/M when compared with miR-294 transfected cells, but Fig. 4d-e shows that it increases the co-expression of G2/M genes. The converse is seen with miR-294 (increased cells in G2/M, but reduced co-expression of G2/M genes). In addition, most of these cell cycle phase genes are not direct targets of let-7c (only 2 out of 36). Therefore, their co-expression is not simply direct targeting of these genes. As discussed in response to reviewer 1, point 1.9, the mechanism underlying this induction of co-expression is something of great interest to us, but we believe beyond the scope of this paper.

3.15) F. Suggested improvements: experiments, data for possible revision

In line 192-193: Can you demonstrate in your experiment that let-7c promotes ESC self-renewal?

Response: Rather than simply repeat the published experiments we refer to in these two lines, we have added expression analysis of pluripotency genes and differentiation genes in the individual cells that received let-7 compared to the other conditions (revised Figure 1d). We confirm that the let-7c retain the undifferentiated pluripotent state, showing no evidence of differentiation.

3.16) H. Clarity and context: lucidity of abstract/summary, appropriateness of abstract, introduction and conclusions

In the title, it is not clear what "microRNAs shape gene co-regulation" means.

We have decided to change the title in response to this comment. The new title is:

The impact of microRNAs on cell heterogeneity and gene co-expression across single embryonic stem cells.

3.17) Overall, in this manuscript the definition of co-regulation is unclear, though this is the central theme. I don't know now if this means that one miRNA regulates many target genes, or that target genes regulate each other because each one acts as a microRNA sponge for the other target genes. This term should be clearly defined in the abstract and in the main text. The conclusions need to be strengthened to show the overall message of the paper, which I feel currently is lacking.

Response: We agree that co-regulation was the incorrect term and we agree that our message/conclusions were not always clearly stated. We have changed co-regulation to co-expression and we define the term early in the text. We are not arguing a sponge effect here as at the introduced and normal physiological levels of these miRNAs, there is unlikely to be a sponge effect, as target sites are saturated (11). Please see our reply to Reviewer 2, point 2.10, for potential causes of co-expression. We have added text throughout

the manuscript to discuss/clarify these and related concepts, which we believe now tightens the conclusions and strengthens the overall message of the paper.

References

1. A. Liberzon *et al.*, The Molecular Signatures Database (MSigDB) hallmark gene set collection. *Cell Syst* **1**, 417-425 (2015).
2. A. Scialdone *et al.*, Computational assignment of cell-cycle stage from single-cell transcriptome data. *Methods* **85**, 54-61 (2015).
3. Y. Wang *et al.*, Embryonic stem cell-specific microRNAs regulate the G1-S transition and promote rapid proliferation. *Nature genetics* **40**, 1478-1483 (2008).
4. Y. Wang *et al.*, miR-294/miR-302 promotes proliferation, suppresses G1-S restriction point, and inhibits ESC differentiation through separable mechanisms. *Cell reports* **4**, 99-109 (2013).
5. C. Melton, R. L. Judson, R. Blelloch, Opposing microRNA families regulate self-renewal in mouse embryonic stem cells. *Nature* **463**, 621-626 (2010).
6. R. M. Kumar *et al.*, Deconstructing transcriptional heterogeneity in pluripotent stem cells. *Nature* **516**, 56-61 (2014).
7. W. T. Guo *et al.*, Suppression of epithelial-mesenchymal transition and apoptotic pathways by miR-294/302 family synergistically blocks let-7-induced silencing of self-renewal in embryonic stem cells. *Cell Death Differ* **22**, 1158-1169 (2015).
8. D. Subramanyam *et al.*, Multiple targets of miR-302 and miR-372 promote reprogramming of human fibroblasts to induced pluripotent stem cells. *Nature biotechnology* **29**, 443-448 (2011).
9. A. M. Klein *et al.*, Droplet barcoding for single-cell transcriptomics applied to embryonic stem cells. *Cell* **161**, 1187-1201 (2015).
10. J. R. Zamudio, T. J. Kelly, P. A. Sharp, Argonaute-bound small RNAs from promoter-proximal RNA polymerase II. *Cell* **156**, 920-934 (2014).
11. A. D. Bosson, J. R. Zamudio, P. A. Sharp, Endogenous miRNA and target concentrations determine susceptibility to potential ceRNA competition. *Mol Cell* **56**, 347-359 (2014).
12. A. Subramanian *et al.*, Gene set enrichment analysis: a knowledge-based approach for interpreting genome-wide expression profiles. *Proc Natl Acad Sci U S A* **102**, 15545-15550 (2005).
13. N. A. Cookson *et al.*, Queueing up for enzymatic processing: correlated signaling through coupled degradation. *Mol Syst Biol* **7**, 561 (2011).

Reviewers' Comments:

Reviewer #1 (Remarks to the Author)

My questions have been answered. Could the authors give a diagrammatic sketch to more clearly illustrate their model?

Reviewer #2 (Remarks to the Author)

The authors addressed my concerns and I now recommend accepting the manuscript.

Reviewer #3 (Remarks to the Author)

The writing of the manuscript has been greatly improved, and is much easier to understand than the first submission. Still, the writing style can be improved further, especially since the point the authors want to make is rather subtle. In particular the fourth paragraph of the Introduction should be rewritten. In line 81, the authors refer two questions, followed by two points in line 81 and 83. But neither of these two points is phrased as a question. It is important at this stage to be very precise in what the authors want to answer. Several reviewers have raised the point that coexpression across single cells between miRNA targets is expected. The authors explain in the rebuttal that this is actually surprising; the same argument is made in the manuscript in the Discussion (line 399 and following). I think this point should be made much earlier, ideally in the Introduction. Now, until the Discussion it may not be clear to the readers what the main point of the manuscript is.

Some smaller comments:

Line 110: This is only true for miR-294, right? If I understand the preceding lines correctly, let-7c transfection did not induce an increase in cell size.

Line 114: Surely mRNA was not converted into cDNA.

Line 159: A comma is needed after let-7c.

Line 159-160: "a seed match to the corresponding to".

Line 191 and 193: This should be 2.2×10^{-16} , not 2.2 times e to the power of -16.

Line 217: "that" -> "than".

Line 262: "effect" should probably be "affect".

Reviewer #1

My questions have been answered. Could the authors give a diagrammatic sketch to more clearly illustrate their model?

Response: We have now added a new Supplementary Fig. 14 (Discussion paragraph, line 398) where we summarize our model.

Reviewer #2

The authors addressed my concerns and I now recommend accepting the manuscript.

Reviewer #3

The writing of the manuscript has been greatly improved, and is much easier to understand than the first submission. Still, the writing style can be improved further, especially since the point the authors want to make is rather subtle. In particular the fourth paragraph of the Introduction should be rewritten. In line 81, the authors refer two questions, followed by two points in line 81 and 83. But neither of these two points is phrased as a question. It is important at this stage to be very precise in what the authors want to answer. Several reviewers have raised the point that coexpression across single cells between miRNA targets is expected. The authors explain in the rebuttal that this is actually surprising; the same argument is made in the manuscript in the Discussion (line 399 and following). I think this point should be made much earlier, ideally in the Introduction. Now, until the Discussion it may not be clear to the readers what the main point of the manuscript is.

Some smaller comments:

Line 110: This is only true for miR-294, right? If I understand the preceding lines correctly, let-7c transfection did not induce an increase in cell size.

Line 114: Surely mRNA was not converted into cDNA.

Line 159: A comma is needed after let-7c.

Line 159-160: "a seed match to the corresponding to".

Line 191 and 193: This should be 2.2×10^{-16} , not 2.2 times e to the power of -16.

Line 217: "that" -> "than".

Line 262: "effect" should probably be "affect".

Response: We have edited the final paragraph in the introduction (lines 65-80) as requested as well as addressed all smaller comments.